# TRPV3 channel activity helps cortical neurons stay active during fever

Yiming Shen[1], Richárd Fiáth[2], Baskar Mohana Krishnan[1], István Ulbert[2,3,4], Michelle W Antoine[1]*

[1]Section on Neural Circuits, National Institute of Alcohol Abuse and Alcoholism, National Institutes of Health, Bethesda, United States; [2]Institute of Cognitive Neuroscience and Psychology, HUN-REN Research Centre for Natural Sciences, Budapest, Hungary; [3]Faculty of Information Technology and Bionics, Pázmány Péter Catholic University, Budapest, Hungary; [4]Center of Neurosurgery and Neurointervention, Semmelweis University, Budapest, Hungary

## eLife Assessment

This is a **valuable** study of the physiological mechanisms promoting network activity during fever in the mouse neocortex. The supporting evidence is **solid**, and has improved with revision, along with increased clarity of presentation.

*For correspondence:
michelle.antoine@nih.gov

**Abstract** Fever raises body temperature ($T_b$) from ~37°C to beyond 38.4°C to combat pathogens. While generally well tolerated below 40°C, in rare cases, fever can abnormally elevate neural activity and induce seizures in neurotypical children aged 2–5 years. This study investigates the mechanisms by which neuronal activity is maintained and stabilized during exposure to fever-range temperatures. Recordings of layer (L)4-evoked spiking in L2/3 pyramidal neurons (PNs) of mouse somatosensory cortex revealed four outcomes as temperature increased from 30°C to 36°C and 39°C (fever-range): neurons remained inactive, stayed active, ceased activity, or initiated activity. Roughly equal proportions of neurons ceased or initiated spiking, making the subset of 'STAY' PNs, those that remain active across temperatures, crucial for maintaining stable cortical output. STAY PNs were more prevalent at younger postnatal ages. Their firing stability was supported by a distinct ion channel composition, including the thermosensitive channel TRPV3, which enables continued spiking by adjusting depolarization to meet spike threshold. Intracellular blockade of TRPV3, but not TRPV4, significantly reduced the proportion of STAY PNs and suppressed spiking at 39°C. Moreover, in $Trpv3^{-/-}$ mice, temperature increases to 39°C reduced both spiking and post-synaptic potential amplitude, and these mice exhibited a delayed seizure onset. Together, these findings suggest that TRPV3 contributes to the preservation of cortical activity during fever.

## Introduction

In most mammals, including humans, core body temperature ($T_b$) is tightly regulated between 36.1 and 37.8°C, with a daily fluctuation of 0.5°C (*Campbell, 2011*). Fever or other thermal challenges can impede heat dissipation to significantly increase $T_b$ beyond 38°C (*Wang et al., 2014*). Because global brain temperature and core body temperature are largely congruent, large increases in temperature affect both regions (*Wang et al., 2014*). Increases in body temperature up to 40°C are generally well tolerated, but temperatures above 42°C (within the hyperthermia and heat stress range) become harmful, leading to protein denaturation and cell death. The body and brain employ protective mechanisms, such as heat shock proteins, to cope with heat stress. These proteins are rapidly induced or

**eLife digest** Fever is a symptom of an infection during which the body temperature rises from just below 37 °C (98.6. 6 °F) to above 38 °C (100.4. 4 °F). This extra heat helps the body fight germs. For most children and adults, a higher temperature – which also affects the brain – causes only milder symptoms such as tiredness, body aches, headache or chills. However, in about 1 in 20 to 1 in 50 children aged 6 months to 5 years, a fever can trigger a seizure. Seizures happen when brain cells, called neurons, become overly active at the same time.

Most studies in rodents have focused on why brain cells become overactive at very high temperatures, around 41 to 42 °C. Much less is known about why seizures are relatively rare at lower temperatures of about 38 to 39 °C. To address this gap, Shen et al. studied how neurons maintain a normal activity at lower temperatures.

The researchers recorded neurons in the somatosensory cortex of mice as their body temperature increased from 30 °C to between 36 °C and 39 °C (fever-range). In young mouse brains, fever induced two simultaneous changes in brain cells: some neurons reduced their activity, allowing them to rest, while others increased their activity to compensate. Because the number of active and less active neurons was roughly balanced, the cells could keep their overall activity stable during a fever.

Moreover, the researchers also identified a thermosensitive protein known as TRPV3 that enhanced its activity during fever. This allowed a sustained flow of ions into the neurons, helping active neurons to keep their firing rate. Rather than increasing seizure risk, this mechanism appears to stabilize neural circuits and prevent excessive synchronization of neurons when fever reduces or pauses activity in many neurons. In addition, active neurons were distinct in that they received greater excitatory input from neighboring neurons and underwent functional adaptations, such as maintaining ion channel expression and distribution, to help preserve effective communication and network stability during fever.

Genetically modified mice lacking TRPV3 showed reduced neuronal activity and a delayed onset of seizures, indicating that TRPV3 is essential in maintaining cortical activity during fever. These findings differ from experiments conducted at very high temperatures, where neuronal overexcitation is driven by breakdowns in ion channel function and inhibitory signaling.

Shen et al. uncovered previously unknown neuronal processes that help the brain continue functioning during fever. This work lays the groundwork for future studies into the causes of fever-related seizures, including whether such seizures arise when compensatory mechanisms fail or when key proteins necessary for maintaining neural balance are overexpressed during elevated body temperatures.

upregulated at noxious temperatures, aiding in the refolding or degradation of damaged proteins once temperatures return to physiological levels (*Miller and Fort, 2018*). However, the mechanisms that stabilize and preserve neuronal firing at lower, fever-range temperatures are not well understood.

In rare cases, moderate-grade fevers ($T_b$ = 38.1 –39°C) cause seizures in children aged 6 months to 6 years, with the highest susceptibility between 2 and 5 years (*Oakley et al., 2009*; *Shinnar and O'Dell, 2004*). A seizure is defined as a sudden, temporary burst of synchronized, excessive neuronal firing in the brain. These febrile seizures (FS) are almost typically triggered by the thermal component of fever, as CNS infections or metabolic disorders are absent (*Shinnar and O'Dell, 2004*). Additionally, FS can be induced solely by increased $T_b$ due to ambient heat, such as a warm bath, hot room, or overdressing. Beyond age five, FS rarely occurs, suggesting that developmental changes in brain physiology stabilize network excitability during this window of susceptibility (*Shinnar and O'Dell, 2004*). Identifying the electrophysiological mechanisms underlying this protective shift could provide insight into normal brain maturation and reveal processes whose disruption may increase susceptibility to febrile seizures.

Rodents exhibit a similar age-dependent susceptibility to thermally induced seizures. The second postnatal week is proposed as a developmental window of heightened seizure susceptibility. During this period, fever-range temperatures (38.1–39°C), comparable to the range at which FSs occur in children (*Holtzman et al., 1981*), do not typically induce seizures. Instead, seizures arise only at higher thresholds (>42.7°C), consistent with hyperthermic rather than febrile seizures. This raises the

question: what physiological changes occur during the second postnatal week that prevent seizures from being triggered at fever-range temperatures (~39°C)?

For practical reasons, EEG recordings for the onset of FS in humans are limited. However, EEG traces from rodents with thermally induced seizures show that ictal events often initiate in the hippocampus and propagate to the cortex, with the later spread to the cortex being correlated to the more severe behavioral stages of the seizure, i.e., Racine stage >4 (*Dubé et al., 2010*; *Baram et al., 1997*). Thus, the cortex represents a critical locus in the seizure phenotype and likely contains mechanisms that may alter neural activity in response to temperature changes. In this manuscript, rather than examining how fever provokes seizures, we focus on identifying the electrophysiological adaptations during postnatal development that stabilize cortical activity at 39°C. Specifically, we compare two developmental stages flanking the proposed window of hyperthermic seizure susceptibility (P12–14): an earlier stage (P7–8) and a later stage (P20–23). This design also provides negative controls, allowing us to determine whether changes observed in the P12–14 window are unique.

Early comparative studies correlating rat and human brain development using neuronal birth dates, myelination patterns, and saltatory growth stages suggest that 5–7 day-old rats approximate full-term human newborns (*Dobbing and Sands, 1973*). More recent analyses based on hippocampal and cortical maturation indicate that the second postnatal week (P10–14) corresponds to human infancy (*Baram et al., 1997*; *Bender et al., 2004*). In contrast, the third and fourth postnatal weeks align with the human juvenile stage (*Baram et al., 1997*). Here, using wild-type mice across three postnatal developmental periods—postnatal day (P)7–8 (newborn), P12-14 (infancy), and P20-26 (juvenile)—we investigated the electrophysiological properties, ex vivo and in vivo, that enable excitatory cortical PNs in mouse primary somatosensory (S1) cortex to remain active during temperature increases from 30°C (standard in electrophysiology studies) to 36°C (physiological temperature), and then to 39°C (fever-range).

Finally, to begin identifying molecular determinants that may mediate the electrophysiological properties enabling firing stability in cortical PNs, we extended our studies to two members of the thermosensitive, $Ca^{2+}$-permeable, nonselective cation channels of the transient receptor potential vanilloid (TRPV) family: TRPV3 and TRPV4. These channels are activated within the innocuous warm temperature range (TRPV3: 31–39°C; TRPV4: 27–34°C) (*Su et al., 2023*; *Shibasaki et al., 2007*), unlike TRPV1 and TRPV2, which respond to noxious heat above 42°C and 52°C, respectively (*Kashio and Tominaga, 2022*). TRPV3 and TRPV4 share low (~41%) amino acid sequence identity (*Smith et al., 2002*) and are both expressed in the brain (*Xu et al., 2002*; *Caterina, 2007*; *Kanju and Liedtke, 2016*; *Chen et al., 2022*). Given its activation within the 31–39°C range, we hypothesized that TRPV3 may specifically contribute to neuronal adaptations occurring within fever range (*Su et al., 2023*; *Shibasaki et al., 2007*).

In summary, we aimed to elucidate adaptative neural mechanisms that allow cortical excitatory PNs to maintain stable spiking despite substantial increases in brain temperature into the fever range. Across three postnatal developmental ages, excitatory cortical PNs responded to depolarizing stimuli during temperature increases from 30°C to 39°C by remaining inactive, continuing to spike, ceasing spiking, or initiating spiking. Nearly equal proportions of neurons ceased or initiated activity, so overall firing stability was maintained largely by 'STAY' neurons, which remained active throughout temperature increases. We propose that STAY neurons likely achieve this stability through unique ion channel compositions, including those from TRPV3, which allows for higher depolarization levels through increased excitatory synaptic input relative to inhibitory input.

## Results

### Body temperature rises to fever levels during regular activity and exposure to higher ambient temperatures

Contrary to common belief, $T_b$ is not fixed but varies with arousal state, motor activity, and environmental temperature (*Kiyatkin, 2019*; *Hankenson et al., 2018*; *Wang et al., 2014*). Using a sterile IPTT-300 implantable temperature transponder (Bio Medic Data Systems, LLC) with a 0.025 s time constant and 0.1°C accuracy, $T_b$ measurements were recorded every 5 min over 6 hr from 2–3 week-old wild-type mice in an open-field arena at room temperature (*Figure 1A*). This analysis revealed a stable median temperature of 36.5°C with a maximum and minimum fluctuation of 0.5°C

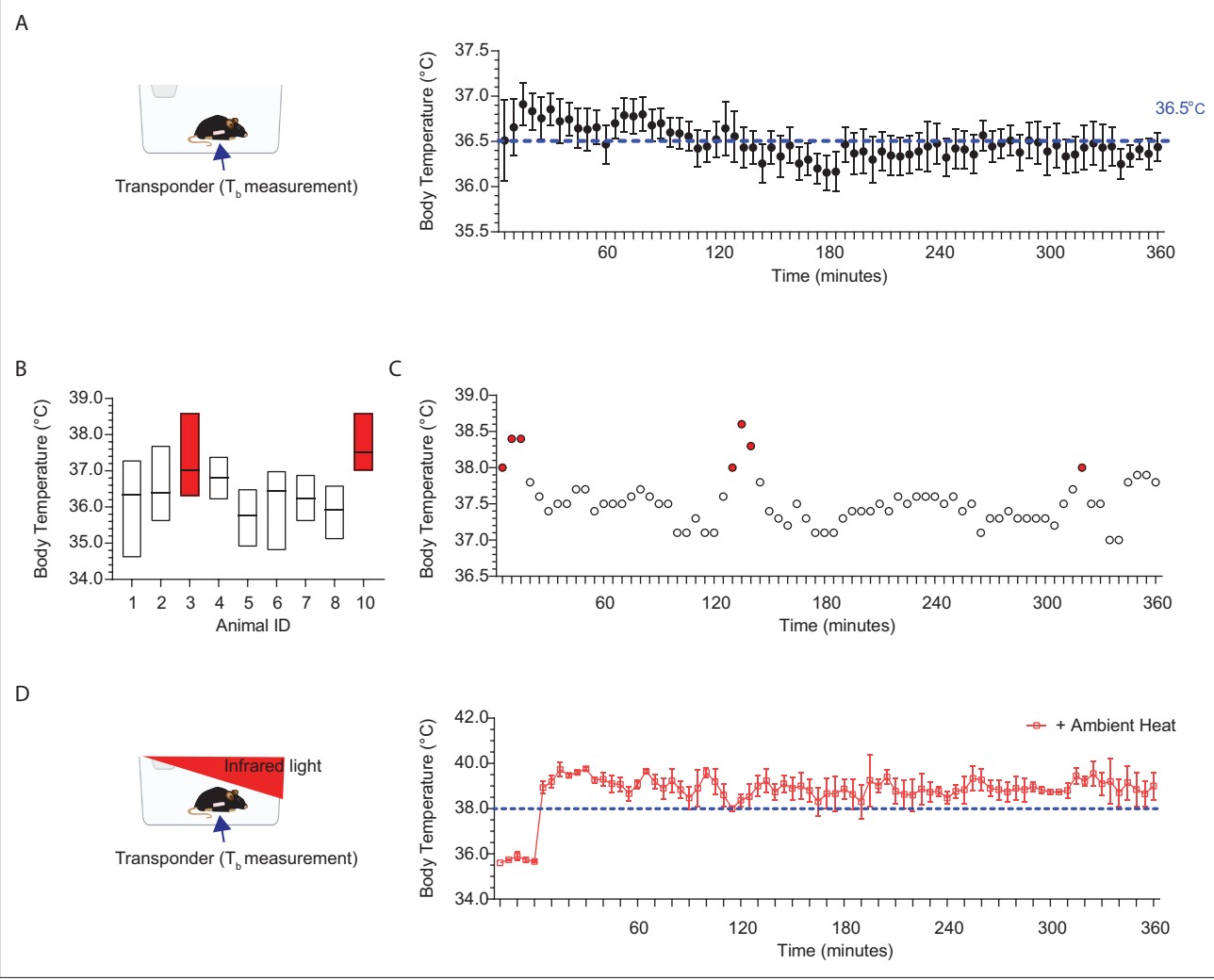

**Figure 1.** Body temperature rises to fever levels during regular activity and upon exposure to higher ambient temperatures. (**A**) Left: Setup for recording mouse body temperature ($T_b$) at room temperature using an implanted transponder for non-invasive measurement. Right: Average $T_b$ readings over 6 hr at 5 min intervals for 10 mice. $T_b$ typically hovers around 36.5°C during the day. (**B**) $T_b$ may briefly elevate into fever range (shaded red bars) during regular daily activity. Median with upper and lower limits indicated. (**C**) $T_b$ readings over 6 hr, at 5 min intervals for animal 3 in **B**. Shaded red dots indicate time points when $T_b$ enters the fever range. (**D**) Left: Setup for $T_b$ recordings during infrared light exposure. Right: $T_b$ elevates into the fever range (>38°C) for extended durations, lasting minutes to hours. Average $T_b$ readings over 6 hr at 5 min intervals for 10 mice.

(*Figure 1A*). However, within-subject analysis showed brief $T_b$ elevations into the fever range (shaded red bars) during increased activity like cage climbing or digging (*Figure 1B–C*). Exposure to infrared light elevates ambient temperatures and induces prolonged $T_b$ elevations into the fever range, with an onset within minutes and a duration of several hours (*Figure 1D*).

## Spiking in P12-P14 cortical excitatory pyramidal neurons remains stable as temperature enters the fever range

To investigate how postnatal cortical neurons respond to fever-range temperature, we used whole-cell current clamp electrophysiology to record natural spiking, evoked synaptically, as brain slice temperature gradually rose from 30°C to 36°C to 39°C in PNs (*Figure 2A–B*). Acute S1 cortex slices (350 µm thick) were prepared using standard methods (*Antoine et al., 2019*). PNs were visually identified via infrared DIC optics, and physiological verification for regular spiking was done in current clamp. Neuronal activity in PNs was evoked by stimulating in the barrel center of cortical layer (L) 4 using a bipolar electrode (0.2 ms pulses) at a specific stimulation intensity, i.e., 1.4X Eθ (*Figure 2A*). Eθ is defined as the minimal intensity evoking an excitatory post-synaptic current (EPSC) during more than

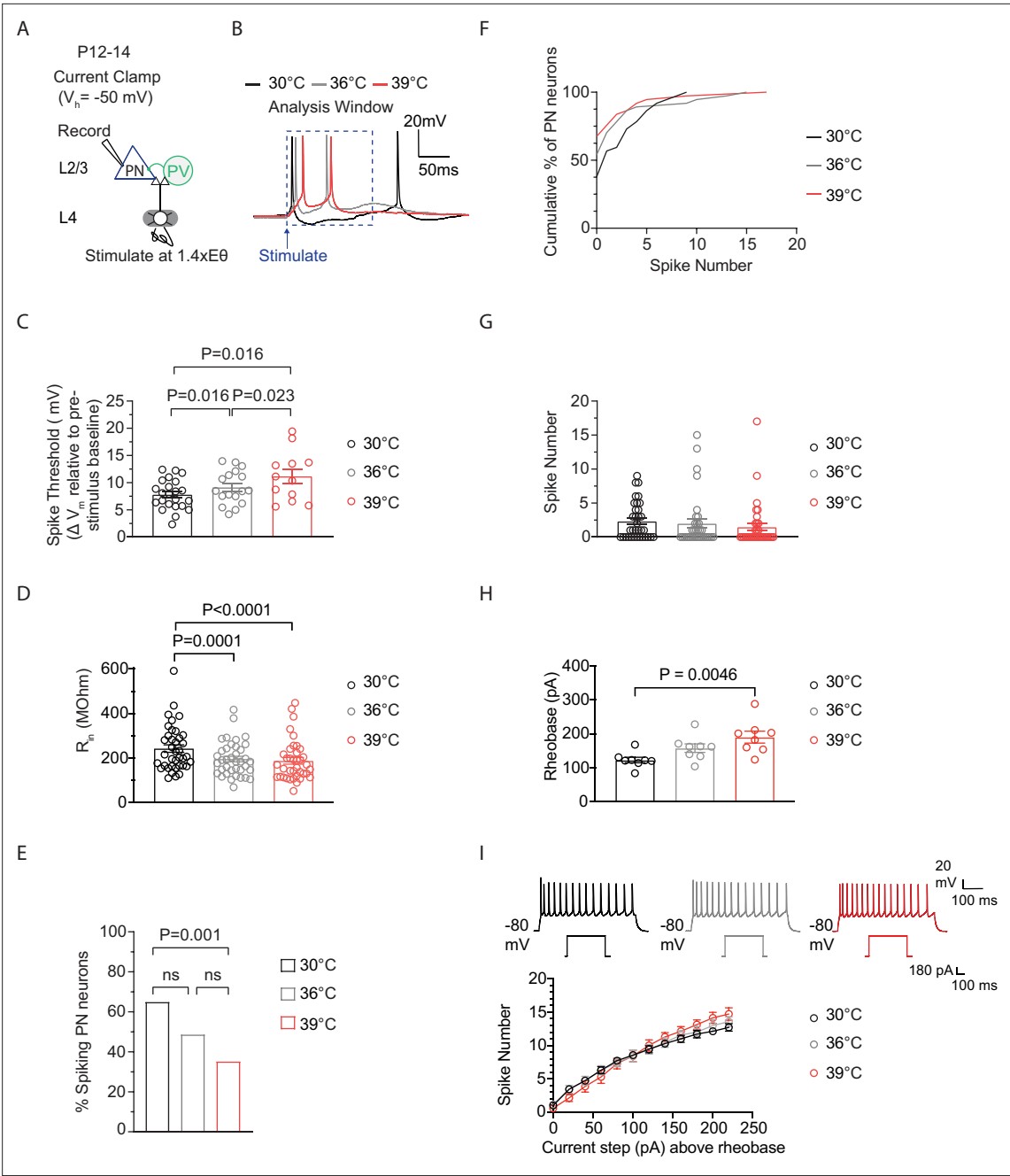

**Figure 2.** Spiking in P12-14 cortical excitatory pyramidal neurons remains stable as temperature enters the fever range. (**A**) Setup for recording L4-evoked post-synaptic potential and spiking in a cortical excitatory pyramidal neuron (PN) at just-subthreshold $V_m$ at 30°C, 36°C, and 39°C in mouse primary somatosensory (S1) cortex. (**B**) Example traces of L4-evoked spikes in L2/3 PN during consecutive recordings at 30°C (black), 36°C (gray), and 39°C (red). (**C**) Depolarization required to reach spike threshold (ST) in L2/3 PNs during temperature elevations from 30°C to 39°C. (**D**) Same as (**C**) for input resistance ($R_{in}$). (**E**) Same as (**C**) for the percentage of spiking L2/3 PNs. (**F**) Same as in (**C**) for the spiking distribution. (**G**) Same as in (**C**) for the number of spikes. (**H**) Rheobase in L2/3 PNs during temperature elevations from 30°C to 39°C. Eight PNs recorded from three animals. (**I**) Quantification of evoked spiking for F-I curves in L2/3 PNs during temperature elevations from 30°C to 39°C. Each recorded cell was exposed to the full temperature range (30– 39°C). Eight PNs recorded from three animals. In **C**, **D** and **G**, each data point represents an individual cell. In C-F, 37 PNs recorded from 14 animals, with each cell exposed to temperature elevations from 30°C to 39°C. Data in **C**, **D**, and **G** are shown as mean ± SEM. Statistical significance was assessed as follows: **C**, **D**, **G**: one-way repeated-measures ANOVA with Tukey's post-hoc test (α=0.05); E – two-tailed binomial test; F – Kolmogorov-Smirnov (K–S) test, (**H**) two-way repeated-measures ANOVA with Tukey's post-hoc test (α=0.05).

3 of 5 consecutive sweeps with 10 s inter-sweep interval. Eθ was determined in voltage clamp for each recorded cell, prior to recording post-synaptic potentials (PSPs) and spiking in current clamp. L4-evoked PSPs and spiking were recorded from L2/3 PNs, with resting membrane potential ($V_m$) set just below spike threshold (–50 mV) to simulate in vivo conditions (*Yamashita et al., 2013*).

Recordings of synaptically-evoked activity in L2/3 wild-type cortical excitatory PNs from P12-14 mice showed significant effects of temperature elevations from 30°C to 39°C. These changes resulted in an increased depolarization required to reach spike threshold (ST) (defined as the minimal $V_m$ that just elicits spiking) (*Figure 2B–C*). Consistent with previous findings, measurements of input resistance ($R_{in}$), which reflect the extent to which membrane channels are open, showed significant reductions across all recorded neurons with increasing temperature (*Figure 2D*). Similar temperature-induced effects on $R_{in}$ have been reported in L2/3 PNs in the rat visual cortex (*Hardingham and Larkman, 1998*) and in hippocampal CA1 and CA3 PNs from P13-P16 mice (*Kim and Connors, 2012*). The depolarized ST and reduced $R_{in}$ in PNs would likely decrease cell excitability and spiking. Indeed, the percentage of spiking cells decreased during temperature elevations from 30°C to 39°C (*Figure 2E*). However, contrary to expectations, average spiking remained remarkably stable (*Figure 2F–G*). We also examined the temperature-induced effects on L2/3 PNs in the absence of excitatory and inhibitory synaptic inputs. Intrinsic spiking was measured in the presence of synaptic blockers (100 µM APV, 10 µM NBQX, and 3 µM gabazine). PNs were held at –80 mV, followed by positive current injection (500 ms) steps to determine rheobase (the minimum current injection needed to produce one spike). At 39°C, rheobase was increased compared to 30°C (*Figure 2H*). Current was injected up to 250 pA above rheobase at 30°C, 36°C, and 39°C, using a new brain slice for each recording. Although rheobase was decreased at 39°C compared to 30°C, we found that the mean number of spikes produced did not differ across current steps (*Figure 2I*).

## Spiking in P7-8 cortical excitatory pyramidal neurons decreases as temperature enters the fever range

This stability in average spiking at 39°C, despite the loss of some previously active neurons, suggests the presence of temperature-adaptive changes in PNs that help to maintain normal circuit activity levels. These changes may either initiate firing in previously non-spiking neurons or either maintain or elevate spiking levels in currently active cells. Interestingly, in P7–8 mice, temperature elevations from 30°C to 39°C produced effects on ST and $R_{in}$ similar to those observed in P12–14 mice, with ST increasing and $R_{in}$ decreasing (*Figure 3A–D*). However, in P7-8 PNs, during temperature elevations from 36°C to 39°C, ST did not increase, and $R_{in}$ decreased (*Figure 3C–D*). Additionally, the percentage of spiking PNs decreased during temperature elevations from 30°C to 39°C (*Figure 3E*). In contrast to P12-14 PNs, the average spiking levels in P7-8 PNs significantly decreased during these temperature elevations (*Figure 3F–G*). This suggests that temperature-adaptive changes, possibly involving $R_{in}$ and ST, which help maintain average spiking levels at fever temperatures despite the loss of some previously active neurons, may be underdeveloped or absent at this earlier age.

## Increased depolarization at higher spike thresholds helps maintain stable spiking activity in cortical pyramidal neurons during temperature elevations

The temperature-induced increases in ST at P12-14 would make it harder for a cell to spike, as larger depolarization levels would be required to maintain spiking (*Figure 4A*). Therefore, we hypothesized that in cells with increased ST that remain spiking, a temperature-adaptive change might involve a proportional increase in depolarization. To determine if larger depolarizations were associated with higher (less negative) STs, we analyzed the relationship between peak post-synaptic potential (PSP), which indicates maximum depolarization, and ST (*Figure 4B*). Correlation analysis revealed a strong positive correlation between peak PSP and spike threshold at P12-14, which was absent at P7-8 and P20-23 (*Figure 4C*). This correlation was specific to these parameters, as it did not occur for comparisons involving input resistance, an intrinsic property which is affected by temperature changes (*Figure 4D–E*).

One mechanism that could promote enhanced depolarization is a reduction in the inhibitory PSP (IPSP) (*Figure 4F*). In addition to the early component of the PSP (the peak), where excitation is maximal, we also quantified the late component, where inhibition levels are greatest and thus

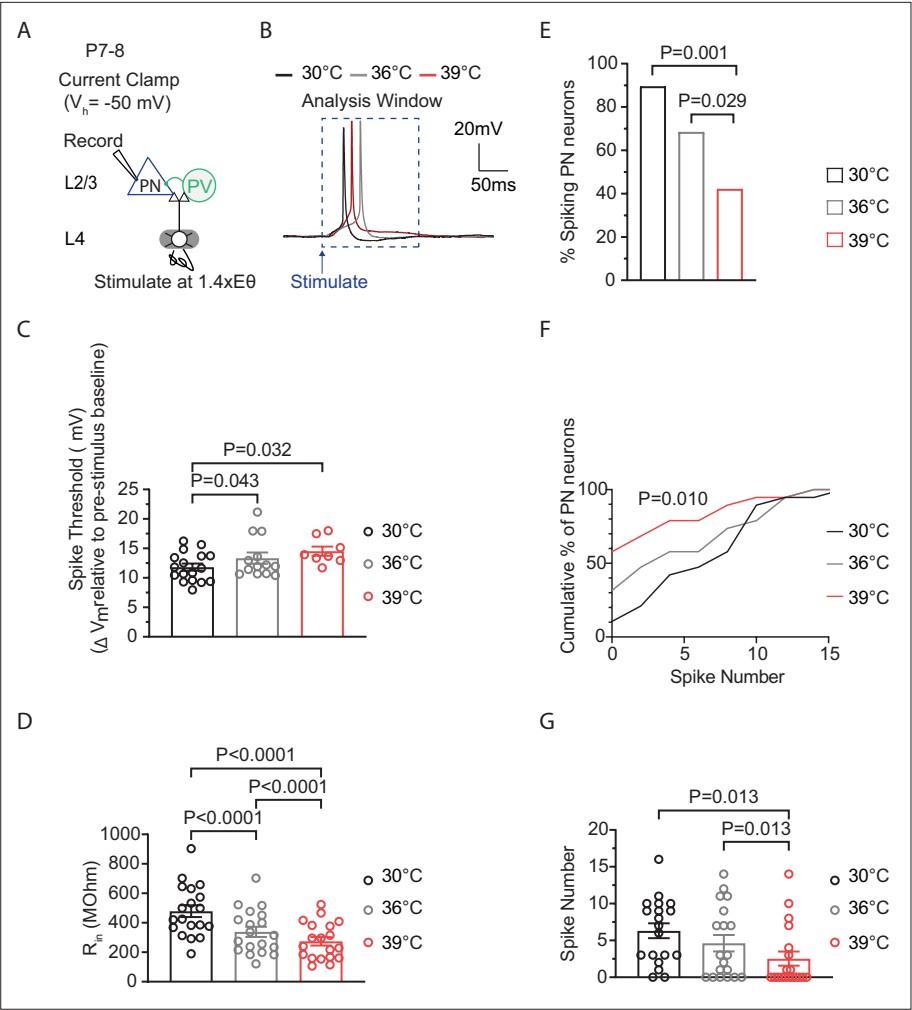

**Figure 3.** Spiking in P7-8 cortical excitatory pyramidal neurons decreases as temperature enters the fever range. (**A**) Setup for recording L4-evoked post-synaptic potential and spiking in an excitatory pyramidal neuron (PN) at just-subthreshold $V_m$ at 30°C, 36°C, and 39°C in mouse primary somatosensory (S1) cortex. (**B**) Example traces of L4-evoked spikes in L2/3 PN during consecutive recordings at 30°C (black), 36°C (gray), and 39°C (red). (**C**) Depolarization required to reach spike threshold (ST) in L2/3 PNs during temperature elevations from 30°C to 39°C. (**D**) Same as (**C**) for input resistance ($R_{in}$). (**E**) Same as (**C**) for the percentage of spiking L2/3 PNs. (**F**) Same as in (**C**) for the spiking distribution. (**G**) Same as in (**C**) for the number of spikes. In **C**, **D**, and **G**, each data point represents an individual cell. In C-G, 19 PNs recorded from five animals, with each cell exposed to temperature elevations from 30°C to 39°C. Data in **C**, **D**, and **G** are shown as mean ± SEM. Statistical significance was assessed as follows: **C**, **D**, **G**: one-way repeated-measures ANOVA with Tukey's post-hoc test (α=0.05); E – two-tailed binomial test; F – Kolmogorov-Smirnov test.

largely represent the IPSP (*Bhatia et al., 2019*; *Antoine et al., 2019*; *Figure 4B*). This analysis revealed that the late PSP (LPSP) was significantly reduced with temperature elevations from 30°C to 39°C (*Figure 4G*). Further analysis indicated that there was no correlation between the ST and LPSP at any age (P7-8: $r=0.17$, $p=0.41$, XY pairs = 25; P12-14: $r=0.24$, $p=0.31$, XY pairs = 19; P20-23: $r=0.33$, $p=0.28$, XY pairs = 13 Pearson's test). However, a significant, moderate relationship occurred between the magnitude of the LPSP and the PSP threshold (*Figure 4H*). Thus, PNs with increased STs and larger PSPs were part of a circuit where synaptic inhibition was reduced across all or most PNs, regardless of whether STs were elevated. These results suggest that at P12-14, ST is sensitive to temperature increases and spiking is maintained by larger depolarization levels in these neurons.

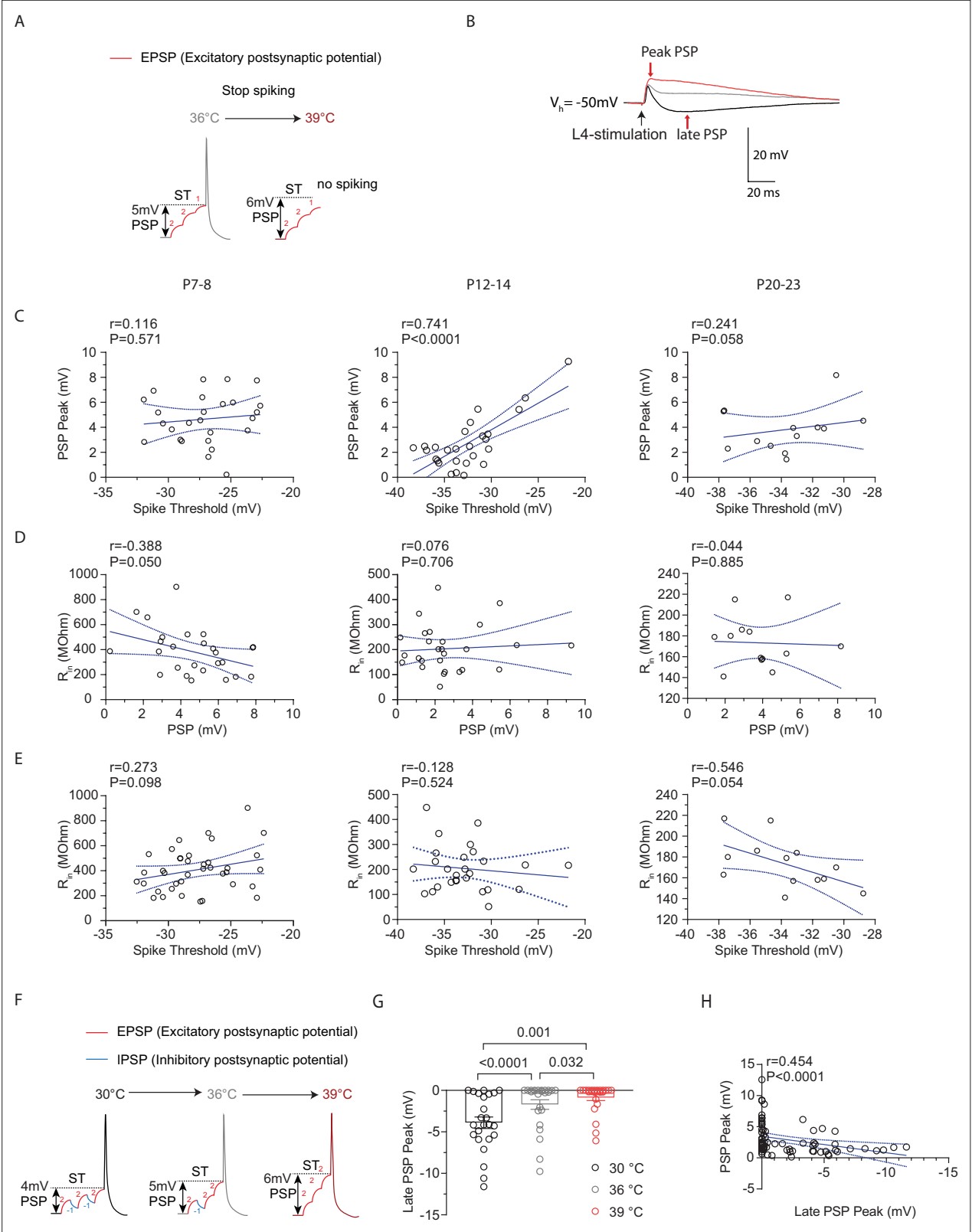

**Figure 4.** Increased depolarization at higher spike thresholds helps maintain stable spiking activity in cortical pyramidal neurons during temperature elevations. (**A**) Illustration demonstrating how temperature-induced changes in spike threshold make spiking more challenging for a neuron, requiring larger levels of depolarization to sustain spiking. (**B**) Example traces of L4-evoked post-synaptic potentials (PSPs) in L2/3 PN during consecutive recordings at 30°C (black), 36°C (gray), and 39°C (red). (**C**) Correlation of PSP peak versus spike threshold (ST). *r*=Pearson correlation coefficient with

*Figure 4 continued on next page*

*Figure 4 continued*

Deming linear regression. (**D**) Correlation of input resistance versus ST. *r*=Pearson correlation coefficient with simple linear regression. (**E**) Same as D but for input resistance versus PSP. (**F**) Illustration showing how temperature-induced loss in inhibition (blue IPSP) could lead to larger levels of depolarization. IPSPs typically bring the membrane potential away from the spike threshold. (**G**) The late PSP peak in L2/3 pyramidal neurons (PNs) during temperature elevations from 30°C to 39°C. Statistical significance was evaluated on log-transformed data using one-way repeated measures ANOVA with Tukey's test, with significance at α=0.05. (**H**) Same as D but for PSP versus Late PSP Peak. *r*=Spearman correlation coefficient with simple linear regression.

## Cortical excitatory neurons that remain active at febrile temperatures exhibit increased spiking rates

At a later age (P24-P26), we applied external heat to elevate normal $T_b$ (36°C) into the fever range and used in vivo high-density, extracellular electrophysiological recordings to capture single-unit activity in the S1 cortex of ketamine/xylazine anesthetized wild-type mice (*Figure 5A*). Unlike in vitro whole-cell patch-clamp recordings, extracellular recordings allow analysis only from cells that spike throughout the recording period during temperature elevations from 36°C (baseline) to fever range. This is largely due to the inability of spike sorting algorithms to reliably track neurons that stop firing during the recording.

Color maps (with yellow and red indicating spiking activity) illustrate multiunit activity obtained in the left and right cortices during baseline temperature (left), warming to fever $T_b$ (middle), and cooling back to 36°C (recovery upon heat removal) are shown in *Figure 5B*. *Figure 5C* presents example traces of spiking activity recorded from three cortical positions (marked by black, gray, and white asterisks on the right side of panels in *Figure 5B*). Following manual curation, single units (n=633) were classified as putative inhibitory interneurons or excitatory PNs based on their spike waveform duration (*Barthó et al., 2004*; see Methods). The spike duration was calculated as the time difference between the trough and the subsequent waveform peak of the mean filtered (300–6000 Hz bandpassed) spike waveform. Durations of extracellularly recorded spikes showed a bimodal distribution (Hartigan's dip test; *p*<0.001) characteristic of the neocortex with shorter durations corresponding to putative interneurons (narrow spikes) and longer durations to putative principal cells (wide spikes). Next, k-means clustering was used to separate the single units into these two groups, which resulted in 140 interneurons (spike duration <0.6 ms) and 493 principal cells (spike duration >0.6 ms), corresponding to a typical 22–78% (interneuron – principal) cell ratio. *Figure 5D* displays representative single units isolated via spike sorting, with unit 3 representing a putative interneuron.

Finally, the firing rates of neurons were computed separately during the three distinguished periods (baseline, thermal fever (39°C), and recovery back to 36°C). To decrease the effect of transient changes (e.g. tissue recovery during the baseline period or the short-term effect of heating during the thermal fever period), for each 45-min-long period, we used only the last 25 min to calculate the mean firing rates of neurons. The normalized average firing rate for each animal (n=5) was calculated for these putative inhibitory interneurons and excitatory PNs across the recording session. We found that in vivo, putative excitatory PNs that remained active increased their spike rates at fever temperatures (*Figure 5E*). The normalized firing rates of five representative units are shown in *Figure 5F* during the baseline, warming (red shaded area), and recovery periods. Additionally, we observed that putative inhibitory interneurons that remained active also increased their spike rates at fever temperatures (*Figure 5E*).

Consistent with these in vivo findings, ex vivo recordings at a similar age (P20-23) showed that 11.1% of recorded excitatory PNs (n=27 cells) spiked at both 36°C and 39°C (*Figure 5G*). However, similar to the in vivo condition, neurons that remained spiking at 36°C and 39°C increased their spike/firing rate (*Figure 5H*). Along with the 11.1% of recorded excitatory PNs that stayed spiking during temperature increases from 36°C to 39°C, roughly equal proportions of neurons stopped spiking and started spiking (*Figure 5G*). Thus, neurons that maintain spiking at fever temperatures play a crucial role in determining the overall activity levels of the circuit. Additionally, we found that the proportion of PNs that remain spiking from 36°C to 39°C is higher at earlier ages (P12-14) compared to later ages (P20-23) (*Figure 5I*).

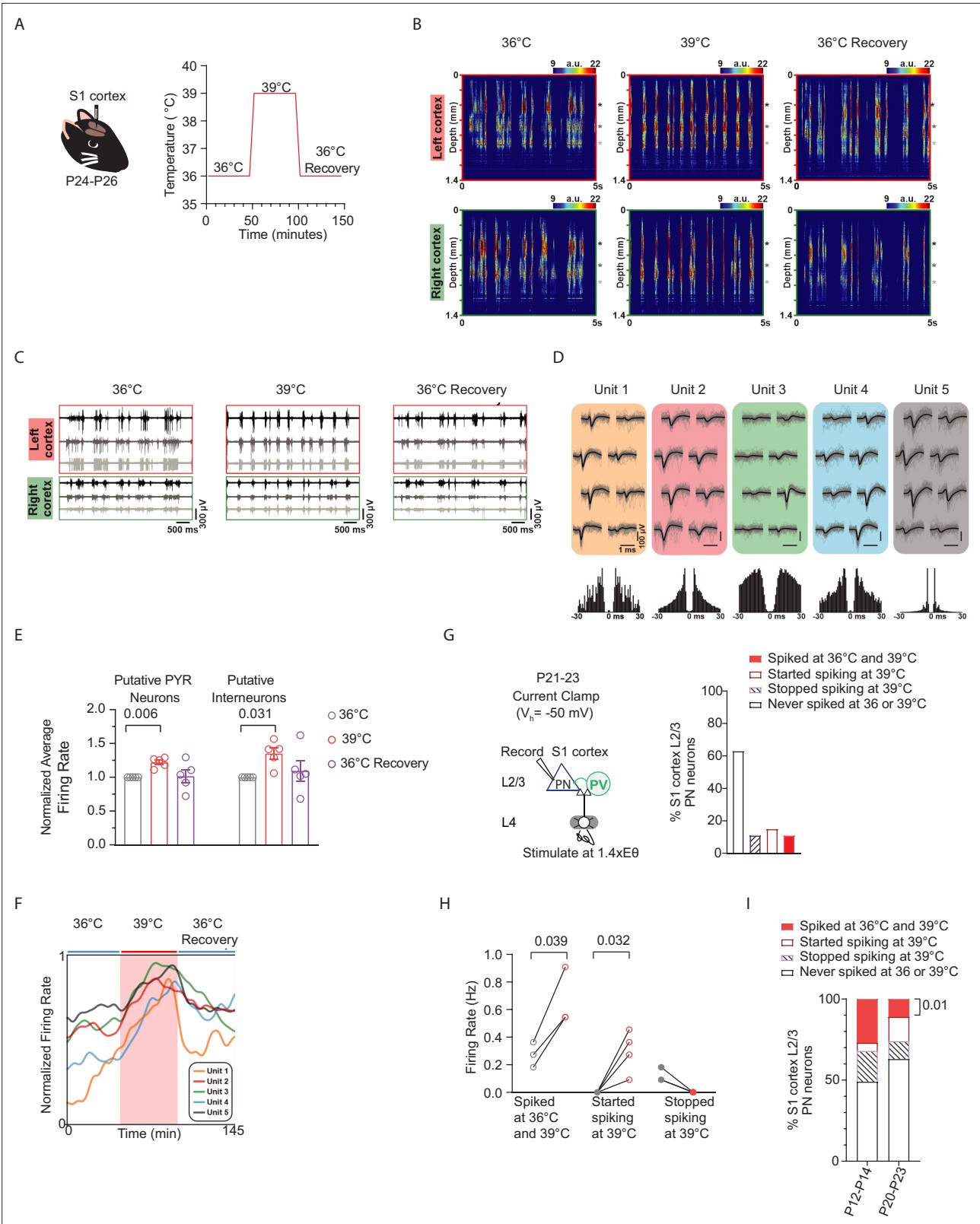

**Figure 5.** Cortical excitatory neurons that remain active at febrile temperatures exhibit increased spiking rates. (**A**) Setup for in vivo recording of spiking in mouse S1 cortex during temperature increases from 36°C to 39°C and cooling back to 36°C. (**B**) Color maps (with yellow and red indicating spiking activity) illustrate multiunit activity obtained in the left and right cortices during baseline temperature (left), warming to fever $T_b$ (middle), and cooling back to 36°C (recovery upon heat removal). (**C**) Example traces of spiking activity recorded from three cortical positions (marked by black, gray, and

*Figure 5 continued on next page*

*Figure 5 continued*

white asterisks on the right side of panels in **B**). (**D**) Mean spike waveforms (black), single spikes (gray), and autocorrelograms (bottom) of single units isolated, with unit 3 representing a putative interneuron. (**E**) Normalized firing rates of putative excitatory pyramidal neurons (PNs) and interneurons obtained from recordings in **A**. (**F**) The normalized firing rates of five representative units during the baseline, warming (red shaded area), and recovery periods. (**F**) presents the same data, with color coding consistent with the units in **D**. (**G**) Left: Setup for ex vivo recording of L4-evoked post-synaptic potentials and spiking in L2/3 cortical PN at just-subthreshold $V_m$ at 30°C, 36°C, and 39°C in mouse S1 cortex. Right: Percent distribution for recorded PNs that never spiked, stopped spiking, stayed spiking, or started spiking during temperature increases from 36°C to 39°C. (**H**) L2/3 PN spiking activity in neurons that stayed spiking (left), started spiking (middle), and stopped spiking (right) upon temperature increases from 36°C to 39°C. (**I**) Activity distribution for ex vivo recorded neurons that never spiked, and those that stopped spiking, stayed spiking, or started spiking upon temperature increases from 36°C to 39°C. *P*-value = 0.01 indicates a significant increase in the fraction of neurons that stayed spiking during temperature increases from 36°C to 39°C (solid red bar). In **E** and **H**, each data point represents an individual animal and cell, respectively. Data were collected from five animals in **B**–**F**, 11 animals in **G**–**H**, and 9–14 animals in **I**. Statistical significance was assessed using one-way repeated-measures ANOVA with Tukey's post-hoc test (**E**), paired two-tailed t-test (**H**), and binomial test (**I**), with significance set at α=0.05.

## Excitatory pyramidal neurons that remain spiking with temperature elevations into fever range exhibit unique intrinsic properties

As neurons that remain spiking with temperature elevations may be crucial in setting overall cortical circuit activity levels, we sought to uncover additional information about the intrinsic characteristics of cortical PNs at P12-14 that facilitate their activity at 36°C to 39°C. We refer to neurons that retained spiking at all three temperatures (30°C, 36°C, and 39°C) as STAY neurons, while those cells that stopped spiking upon temperature transitions from 30°C to 36°C or 36°C to 39°C are denoted as STOP neurons. Firstly, in STOP and STAY neurons, ST increased during temperature elevations from 30°C to 39°C (*Figure 6A*). However, in STOP PNs, the increases in ST were greater than in those neurons that continued spiking (*Figure 6A*). The ST of neurons that continued spiking at 39°C was similar in magnitude to that of neurons that stopped spiking at 36°C.

There was an interaction of temperature and whether the PNs were STOP or STAY cells in the PSP (*p*=0.048, two-way ANOVA, mixed-effects model), with a trend toward larger PSPs in the STAY PNs (*Figure 6B*). STAY cells showed a strong positive correlation between PSP and ST (*Figure 6C–D*). At 30°C, STAY neurons had larger LPSPs (*Figure 6E*). However, inhibition levels in these PNs were unique in that inhibition levels significantly decreased with temperature elevations (*Figure 6E*). Moreover, we found a significant moderate inverse correlation between ST and LPSP in the STAY cells that remain spiking at 39°C (*r*=−0.57, *p*=0.046, XY pairs = 13, Spearman). Thus, in PNs that stay spiking, the excitation and inhibition levels are matched to facilitate greater depolarization and maintain spiking activity. Temperature-induced alterations in spike height and spike afterhyperpolarization (AHP) were not different between STAY and STOP cells, while $R_{in}$ was largely insensitive to temperature changes in STAY PNs but not in STOP PNs (*Figure 6F–H*).

## Temperature elevations in the fever range increase TRPV3 currents in cortical pyramidal neurons

Next, we sought to identify molecular determinants that might underlie the properties of STAY cells. We focused on the transient receptor potential vanilloid 1–4 (TRPV1-4) channels, a family of thermo-sensitive, $Ca^{2+}$-permeable, nonselective cation channels. Of these, only TRPV3 and TRPV4 are activated within the innocuous warm temperature range (31–39°C), whereas TRPV1 and TRPV2 respond to noxious heat above 42°C and 52°C, respectively (*Kashio and Tominaga, 2022*). TRPV3, in particular, has a reported temperature threshold for activation between 31–39°C (*Smith et al., 2002*), noting activation at 39°C.

We examined TRPV3 protein expression by immunohistochemical staining at P7, P14, and P21 (*Figure 7A–B*). Staining specificity was confirmed at all ages by co-incubating the TRPV3 antibody with a TRPV3 blocking peptide, with an example shown for a P7 brain section (*Figure 7A*, right panel). TRPV3 expression was detected in S1 cortex at all ages, with strong expression in L2/3 at P14 but not at P7 or P21 (*Figure 7A*). Outside the S1 cortex, TRPV3 was also present in the hippocampus, thalamus, and striatum (*Figure 7A–B*). Co-immunostaining for TRPV4 revealed several brain regions where TRPV4 (red color) and TRPV3 (green color) were non-overlapping (*Figure 7B*, top-left panel). In the S1 cortex at P14, TRPV3 was detected in L4 barrels and showed a patchy distribution in L2/3, with

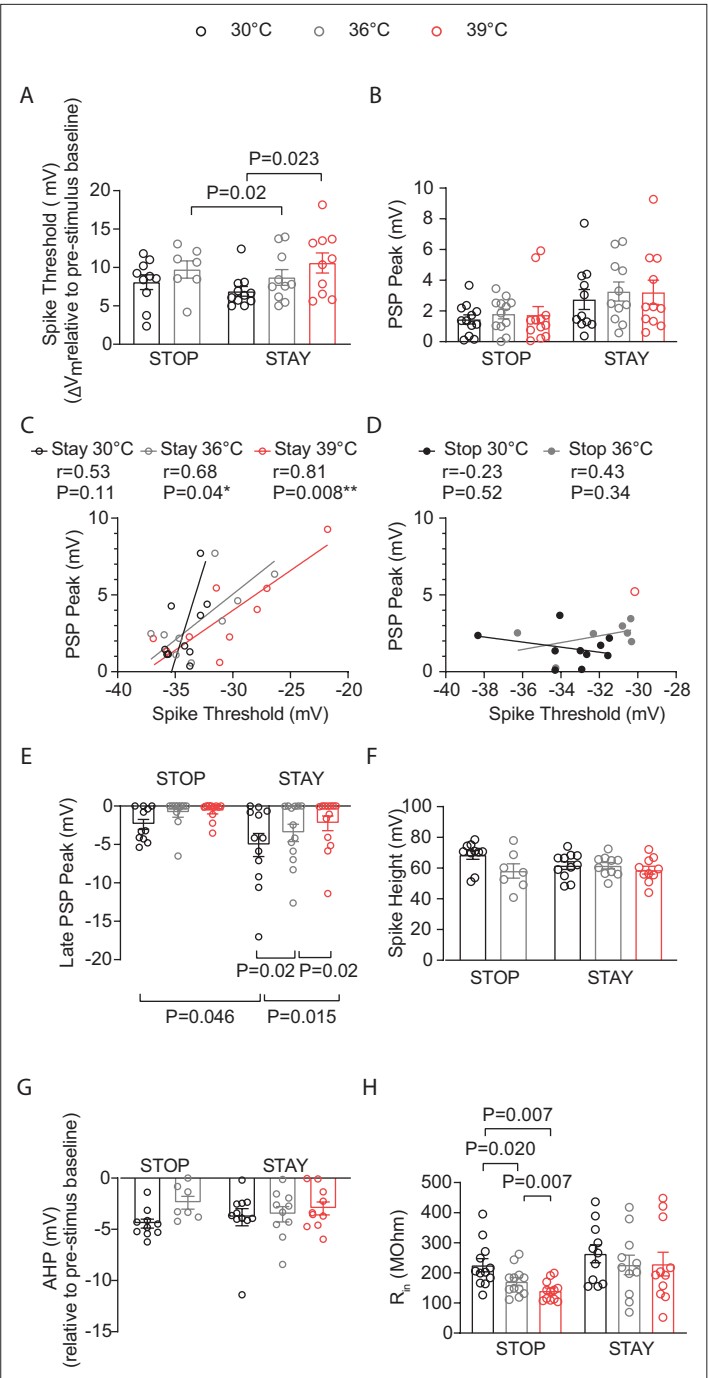

**Figure 6.** Excitatory pyramidal neurons that remain spiking with temperature elevations into fever range exhibit unique intrinsic properties. Neurons that spiked at all temperatures (30°C, 36°C, and 39°C) are STAY pyramidal neurons (PNs), while those that stopped spiking at 36°C or 39°C are STOP neurons. (**A**) Depolarization required to reach spike threshold (ST) in STOP and STAY L2/3 PNs during temperature elevations from 30°C to 39°C. (**B**) Same as (**A**) for L4-evoked post-synaptic potentials (PSPs). (**C**) Correlation of PSP peak versus ST in STAY L2/3 PNs at 30°C, 36°C, and 39°C. *r*=Pearson correlation coefficient with Deming linear regression. (**D**) Same as (**C**) for STOP cells. (**E**) Same as (**A**) for the L4-evoked late PSP peak. (**F**) Same as (**A**) for spike height. (**G**) Same as (**A**) for spike afterhyperpolarization (AHP). (**H**) Same as (**A**) for input resistance. Each data point in **A**–**H** represents an individual cell. Data were collected from 14 animals. Mean ± SEM is shown in **A**–**B** and **E**–**H**. Statistical significance was assessed using one- or two-way repeated-measures ANOVA with Tukey's or Sidak post-hoc test (**A**–**B**, **E**–**H**; α=0.05). In **C**–**D**, correlations were evaluated using Pearson's r with Deming linear regression.

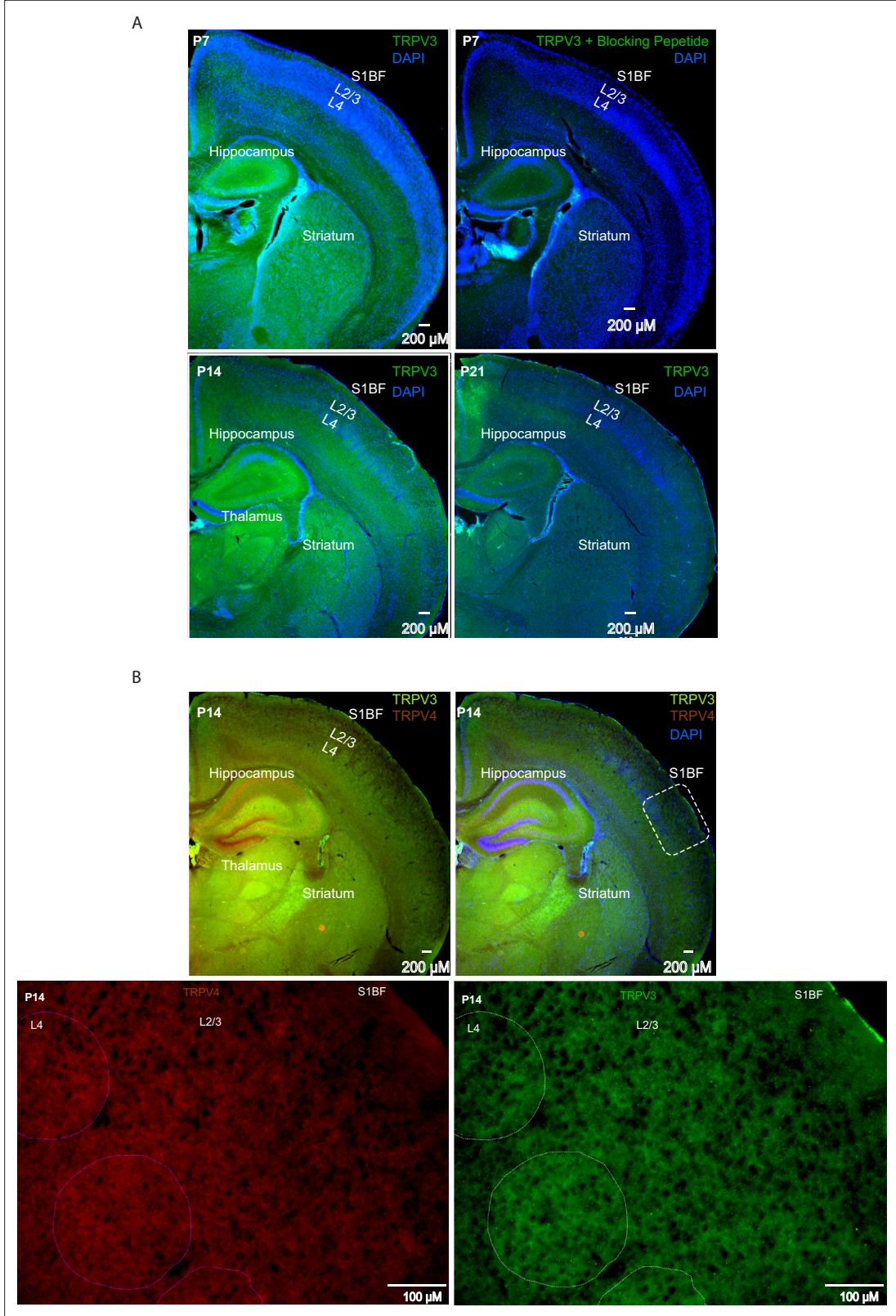

**Figure 7.** TRPV3 protein expression by immunohistochemistry at postnatal days 7, 14, and 21. (**A**) TRPV3 immunostaining (green) in mouse primary somatosensory (S1) cortex, striatum, thalamus, and hippocampus, with DAPI counterstain (blue). Top row left: Postnatal day (P)7; bottom row left: P14; bottom row right: P21. Top row right: Sections incubated with TRPV3 antibody plus TRPV3 blocking peptide in S1 cortex, striatum, thalamus, and hippocampus (green) with DAPI counterstain (blue). Remaining TRPV3 staining is shown in green with DAPI in blue. (**B**) TRPV3 and TRPV4

*Figure 7 continued on next page*

*Figure 7 continued*

immunostaining in S1 cortex, striatum, thalamus, and hippocampus at P14. Top row left: TRPV3 (green) and TRPV4 (brick red). Top row right: TRPV3 (green) and TRPV4 (brick red) with DAPI counterstain (blue). Bottom row left: TRPV4 immunostaining in S1BF (inset from top row left) at 10x. Bottom row right: TRPV3 immunostaining in S1BF (inset from top row left) at 10x.

some patches displaying more intense staining (*Figure 7B*, bottom-right panel). This pattern was not observed for TRPV4 (*Figure 7B*, bottom-left panel).

Using whole-cell voltage clamp, we recorded TRPV3 currents at different voltages in cortical excitatory PNs with bath application of camphor (5 mM), a TRPV3 agonist (*Moqrich et al., 2005*; *Figure 8A*). We observed a significant increase in TRPV3 currents at 39°C (*Figure 8B–C*, two-way repeated measures ANOVA, with Fisher's LCD test). We also measured the net TRPV3 currents by comparing TRPV3 current densities recorded in the absence and presence of forsythoside B, a natural TRPV3 inhibitor found in the plant *Lamiophlomis rotate* with an IC 50 of 6.7 μmol/L and no obvious inhibitory effects on the other TRPV channels like TRPV1 or 4 (*Zhang et al., 2019*). Consistent with *Smith et al., 2002*, who noted TRPV3 activation at 39°C, we also observed a net positive TRPV3 current at 39°C (*Figure 8D–G*).

## Inhibiting TRPV3, but not TRPV4 channels, significantly reduced the population of STAY pyramidal neurons and spiking levels at fever temperature

To investigate the potential role of TRPV3 and TRPV4 in regulating STAY PN phenotype in the S1 cortex during temperature increases from 30–39°C, we applied the following blockers: forsythoside B (50 μM) or RN1734 (10 μM, TRPV4 blocker) intracellularly in our recordings of L4-evoked spiking at 1.4X Eθ in L2/3 PNs (*Figure 9A*). RN1734 is a highly selective antagonist for TRPV4 with an IC 50 of 5.9 μmol/L and minimal to negligible affinity for the other TRPV channels (*Vincent et al., 2009*). These blockers phenocopy electrophysiological effects observed in *Trpv4*[-/-] KO and *Trpv3*[-/-] KO mice

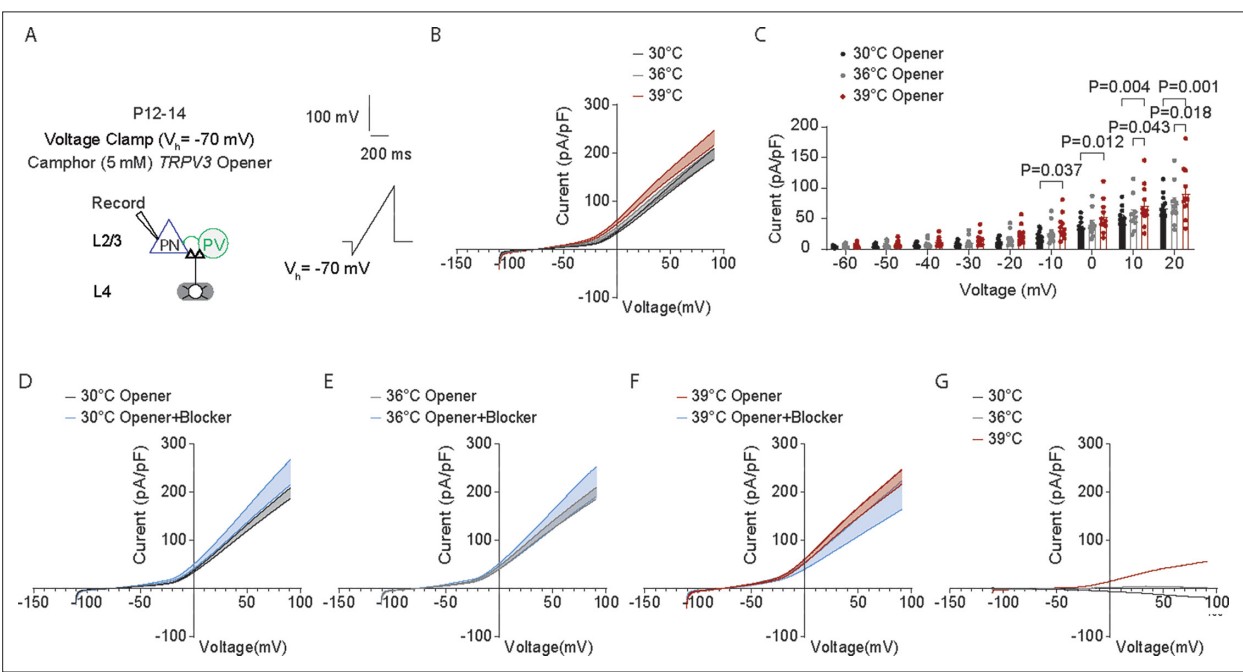

**Figure 8.** Temperature elevations in fever range increase TRPV3 currents in cortical pyramidal neurons. (**A**) Setup for recording whole-cell TRPV3 currents at 30°C (black), 36°C (gray), and 39°C (red) in cortical excitatory pyramidal neurons (PNs) with bath application of camphor (5 mM), a *TPRV3* agonist. (**B**) Current density-voltage (I–V) relationship of TRPV3 currents at 30°C (black), 36°C (gray) and 39°C (red) in wild-type (WT) mice: 11 cells from four mice. (**C**) Scatter dot plots of the current density-voltage measurements. (**D**) Current density-voltage (I–V) relationship of TRPV3 currents at 30°C (black) in the presence of camphor (5 mM), a TPRV3 agonist, or camphor (5 mM) + TRPV3 blocker (Forsythoside B, 50 μM) (blue). (**E**) Same as (**D**) but for 36°C. (**F**) Same as (**D**) but for 39°C. (**G**) Current density-voltage (I–V) plot showing the net TRPV3 current (opener – (opener+blocker) condition). In **B-F**, statistical significance was assessed using a two-way repeated-measures ANOVA with Tukey's or Sidak post-hoc test (α=0.05).

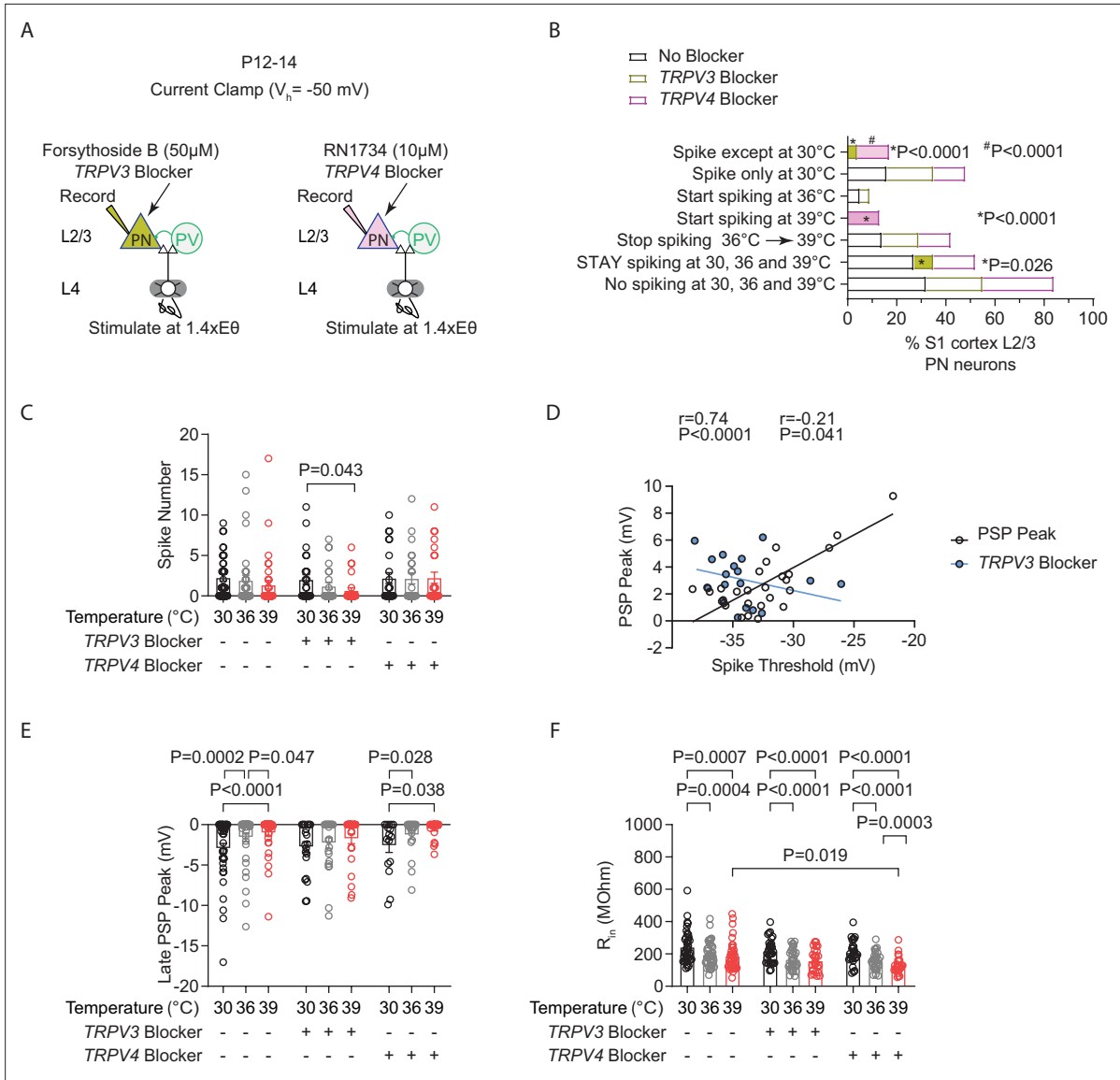

**Figure 9.** Inhibiting TRPV3, but not TRPV4 channels, significantly reduced the population of STAY pyramidal neurons and spiking levels at fever temperature. (**A**) Setup for recording L4-evoked post-synaptic potential and spiking in an excitatory cortical pyramidal neuron (PN) with an intracellular blocker of TRPV3 channels (Forsythoside B, 50 μM) (left) or TRPV4 channels (RN1734, 10 μM) (right) at just-subthreshold $V_m$ at 30°C, 36°C, and 39°C in mouse S1 cortex. (**B**) Percentages of cell types obtained from experiment in **A**. (**C**) Evoked spikes in L2/3 cortical PNs during temperature elevations to 30°C, 36°C, and 39°C under three conditions: no blockers, TRPV3 blocker (Forsythoside B, 50 μM), or TRPV4 blocker (RN1734, 10 μM). (**D**) Correlation between post-synaptic potential (PSP) peak and spike threshold (ST). *r*=Pearson correlation coefficient with Deming linear regression. (**E**) Same as (**C**) for the L4-evoked late PSP peak. (**F**) Same as (**C**) for input resistance ($R_{in}$). Each data point in **C–F** represents an individual cell. Data were collected from 26 cells in 7 animals for the TRPV3 blocker, 24 cells in 6 animals for the TRPV4 blocker, and 37 cells in 14 animals for the no-block condition. Mean ± SEM is shown in **C**, **E**, and **F**. Statistical significance was assessed using one- or two-way repeated-measures ANOVA with Tukey's or Sidak post-hoc test (α=0.05). In **D**, correlations were evaluated using Pearson's r with Deming linear regression.

(***Chen et al., 2022***). For instance, *Trpv3*$^{-/-}$ KO mice exhibit a hyperpolarizing effect on the resting membrane potential (RMP) in striatal MSNs and L2 stellate neurons in the entorhinal cortex (***Chen et al., 2022***). This hyperpolarization reflects reduced TRPV3 activity, which normally promotes cation influx (***Shibasaki et al., 2007***; ***Chen et al., 2022***). Although increases in temperature alone can depolarize the RMP, consistent with prior reports in *Trpv3*$^{-/-}$ mice, our recordings of L4-evoked spiking in L2/3 PNs of S1 cortex showed a similar trend toward RMP hyperpolarization with intracellular TRPV3 block via forsythoside at 30°C and 36°C, and a significant enhancement at 39°C. Specifically, in the absence of block (n=13 cells) versus with TRPV3 blocker (n=15 cells), the RMP values were:

30°C: −75.1±1.1 vs. −77.1±1.0 mV (*p*=0.273); 36°C: −73.5±1.2 vs. −76.3±1.1 mV (*p*=0.273); 39°C: −71.3±1.0 vs. −74.4±1.1 mV (*p*=0.048); two-way repeated-measures ANOVA with Fisher's LSD test.

Furthermore, we found that intracellular blockade of TRPV3 caused a significant reduction in the percentage of STAY neurons (No block: 27% (37 recorded cells), TRPV3 block: 8% (n=26 recorded cells), TRPV4 block: 17% (n=24 recorded cells), two-tailed binomial test) (*Figure 9B*). Consistent with the STAY cells being largely responsible for maintaining stable average spiking during temperature elevations, spiking was reduced in PNs with intracellular TRPV3 blockade during the temperature transitions from 30°C to 39°C (*Figure 9C*). TRPV3 block did not occlude the increases in the depolarization required to reach ST (TRPV3 blocker: 30°C: 7.15±0.57, 36°C: 8.39±0.26, 39°C: 10.0±0.94, 30°C vs 39°C: *p*=0.007, One-way repeated measures ANOVA, mixed-effects model with Tukey's test). However, it did abolish the temperature-induced increases in the PSP (TRPV3 blocker: 30°C: 2.37±0.36, 36°C: 2.45±0.38, 39°C: 2.22±0.25; No blocker: 30°C: 2.47±0.32, 36°C: 2.90±0.32, 39°C: 3.32±0.52, 30 vs 39°C: *p*=0.030, One-way repeated measures ANOVA, mixed-effects model with Dunnett's test). Consistently, TRPV3 block abolished the positive correlation between ST and PSP (No block: XY pairs = 27, TRPV3 blocker: XY pairs = 18, Pearson's test) (*Figure 9D*).

Furthermore, in keeping with the correlation between the magnitude of the LPSP and the PSP threshold, we found that TRPV3 blockade also prevented the temperature-induced decreases in the LPSP (*Figure 9E*). Moreover, these reductions in spiking with TRPV3 blockade were not dependent on temperature-induced decreases in $R_{in}$, as changes to PNs were similar in the TRPV3 block and no-block conditions (*Figure 9F*). Altogether, these results suggest that TRPV3 expression or its functional effects—such as mediating $Ca^{2+}$ or other cation ($Na^+$, $K^+$) influx to promote increased depolarization with spike threshold increases—may define the subset of STAY neurons. Given this role, the loss of TRPV3 is expected to promote temperature-induced hypoactivity in cortical circuits rather than exacerbate seizure activity.

## TRPV3 knockout mice exhibit reduced spiking activity at febrile temperatures and delayed seizure onset

Recordings of L4-evoked PSPs and spiking in L2/3 cortical PNs from wild-type (WT) and homozygous *Trpv3* knockout (*Trpv3⁻ᐟ⁻*) (*Moussaieff et al., 2008*) mice revealed similar degrees of temperature sensitivity in ST and $R_{in}$. Febrile temperatures increased ST and decreased $R_{in}$ (*Figure 10A–C*). Consistent with findings that TRPV3 blockade using 50 µM forsythoside B reduces spiking in cortical L2/3 PNs, we observed significantly reduced spiking in *Trpv3⁻ᐟ⁻* mice as well (*Figure 10D*). Analysis of post-synaptic potentials in these neurons showed that, in WT mice, PSP amplitude increased with temperature elevation into the febrile range, whereas this temperature-dependent depolarization was absent in *Trpv3⁻ᐟ⁻* mice (*Figure 10E*). This reduction in spiking is consistent with the idea that, during increases in STs, concurrent increases in PSP amplitude help sustain neuronal spiking. These findings support the conclusion that TRPV3 contributes to enhanced depolarization and excitability at febrile temperatures.

We next examined the effects of reduced *Trpv3* expression on hyperthermia-induced seizures in WT (*Trpv3⁺ᐟ⁺*), heterozygous (*Trpv3⁺ᐟ⁻*), and homozygous knockout (*Trpv3⁻ᐟ⁻*) P12 pups. As described in *Figure 1D*, mice were exposed to infrared light to raise their core $T_b$ into the range of hyperthermia (>40°C; *Figure 10F*). To enable wireless, non-invasive $T_b$ recordings, all mice were implanted with temperature transponders. Baseline $T_b$ measurements prior to infrared exposure revealed no significant differences across genotypes (*Trpv3⁺ᐟ⁺*: 33.7 ± 0.44°C, *Trpv3⁺ᐟ⁻*: 33.7 ± 0.33°C, *Trpv3⁻ᐟ⁻*: 32.8 ± 0.10°C; One-way ANOVA with Tukey's multiple comparison test, F (2,11)=2.41, *p*=0.136). As $T_b$ increased during infrared exposure, P12 mice typically transitioned from rest to active walking and exploratory behavior. The $T_b$ at which mice became ambulatory did not significantly differ across genotypes (*Trpv3⁺ᐟ⁺*: 37.5 ± 0.55°C, *Trpv3⁺ᐟ⁻*: 36.3 ± 0.43°C, *Trpv3⁻ᐟ⁻*: 36.6 ± 0.34°C; One-way ANOVA with Tukey's multiple comparison test, F(2,11) = 1.34, *p*=0.301).

As $T_b$ reached 41–42°C, mice exhibited loss of postural control (LPC), defined as collapse and failure to maintain upright posture. LPC is a key behavioral marker of cortical and motor circuit dysfunction. The $T_b$ at which LPC occurred was significantly lower in *Trpv3⁻ᐟ⁻* mice compared to wild-type controls (*Figure 10G*). LPC was followed by the rapid onset of generalized seizures. While the $T_b$ at seizure onset was not significantly different among genotypes (*Figure 10H*), the time to seizure onset was significantly prolonged in both *Trpv3⁺ᐟ⁻* and *Trpv3⁻ᐟ⁻* mice. Specifically, *Trpv3⁺ᐟ⁻* mice exhibited a 49.4%

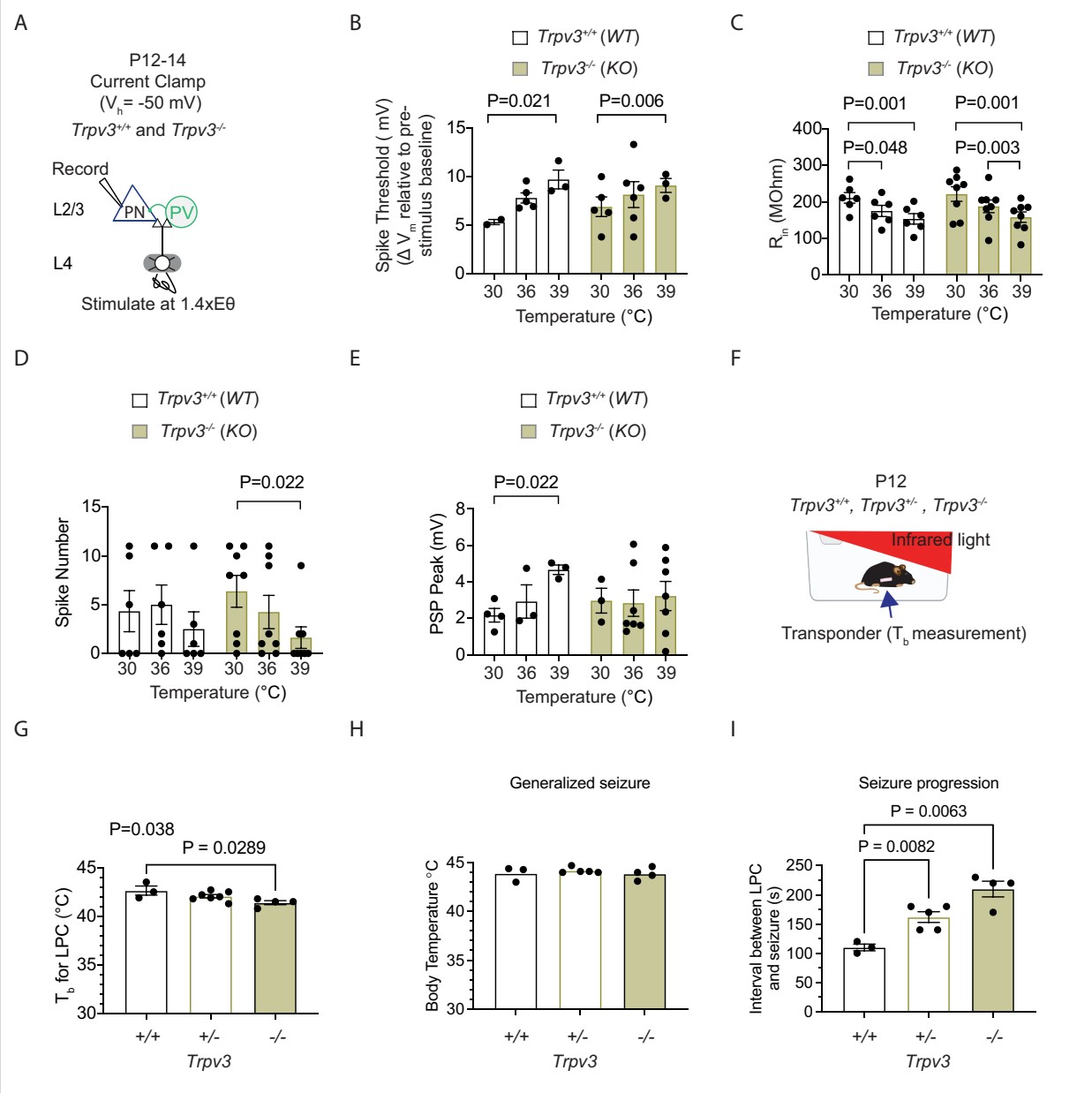

**Figure 10.** TRPV3 knockout mice exhibit reduced spiking activity at febrile temperatures and delayed seizure onset. (**A**) Setup for recording L4-evoked post-synaptic potentials and spiking in excitatory cortical pyramidal neurons (PNs) at just-subthreshold Vm at 30°C, 36°C, and 39°C in mouse S1 cortex of wild-type (*Trpv3$^{+/+}$*) and Trpv3 knockout (*Trpv3$^{-/-}$*) mice. (**B**) Depolarization required to reach spike threshold (ST) in *Trpv3$^{+/+}$* and *Trpv3$^{-/-}$* mice at 30°C, 36°C, and 39°C. (**C**) Same as (**B**) but for input resistance (R$_{in}$). (**D**) Same as (**B**) but for number of spikes. (**E**) Same as (**B**) but for post-synaptic potential (PSP). (**F**) Setup for recording mouse body temperature (T$_b$) at room temperature and during fever-range and higher, using an implanted transponder for non-invasive measurement, with exposure to infrared light. (**G**) Time to loss of postural control (LPC), defined as collapse and failure to maintain upright posture, in wild-type (*Trpv3$^{+/+}$*), heterozygous (*Trpv3$^{+/-}$*), and Trpv3 knockout (*Trpv3$^{-/-}$*) mice. (**H**) Same as in (**G**) but showing T$_b$ at seizure onset. (**I**) Same as in (**G**) but for the time from LPC to seizure onset. Each data point in **B**–**E** represents an individual cell (three animals per genotype). Statistical significance was assessed using two-way repeated-measures ANOVA with Tukey's or Sidak post-hoc test (α=0.05). Each data point in **G**–**I** represents an individual animal. Statistical significance was assessed using one-way repeated-measures ANOVA with Tukey's or Sidak post-hoc test (α=0.05).

increase, and *Trpv3$^{-/-}$* mice a 91% increase in latency to seizure compared to WT mice (*Figure 10I*). Importantly, the rate of T$_b$ increase leading to seizure did not differ across genotypes (*Trpv3$^{+/+}$*: 0.011±0.003°C/s, *Trpv3$^{+/-}$*: 0.013±0.002°C/s, *Trpv3$^{-/-}$*: 0.012±0.001°C/s; One-way ANOVA with Tukey's multiple comparison test, F(2,11) = 0.1055, *p*=0.9008), suggesting the delayed seizure onset was not due to slower heating. Together, these findings indicate that reduced TRPV3 function increases

resistance to seizure initiation and/or propagation under febrile conditions, likely by decreasing neuronal excitability.

## Discussion

In this study, we aimed to elucidate adaptive neural mechanisms that enable excitatory PNs to remain active despite substantial increases in brain temperature into the fever range. Our electrophysiological assessments across three postnatal ages, P7-8, P12-14, and P20-23, showed that excitatory PNs responded to temperature increases from 30°C to 39°C by either remaining inactive, staying active, ceasing activity, or initiating activity. Roughly equal portions of neurons ceased or initiated activity, so firing stability was largely maintained by those that remained active, i.e., STAY neurons. The overall proportion of PNs that remain spiking from 36°C to 39°C was higher at P12-14 compared to P20-23. This increased proportion at earlier ages could potentially reflect the increased risk for febrile seizure occurrence at earlier postnatal ages.

STAY neurons likely possess a unique composition of ion channels that enable them to exhibit temperature sensitivity in spike threshold but not in input resistance and adjust their depolarization levels to match the ST value. Our initial characterization focused on *Trpv3* and *Trpv4*, two members of a family of thermosensitive transient receptor potential vanilloid channels that are expressed in the brain and activated between 30–39°C, the temperature range of our study. Intracellular blockade of TRPV3, but not TRPV4, significantly decreased the population of STAY PNs and prevented stability in spiking activity during temperature increases into fever range.

TRPV3 is highly expressed in various tissues, including epithelial cells of the skin and oropharynx, the tongue, testis, dorsal root ganglion, trigeminal ganglion, spinal cord, and brain (Moussaieff A et al., 2008 and *Xu et al., 2002*). It plays a crucial role in skin and hair generation as well as in the perception of pain and warmth on the skin. However, its role in the central nervous system remains to be further elucidated. In the brain, TRPV3 is expressed in several neuronal types, including cortical and striatal neurons (*Xu et al., 2002*; *Chen et al., 2022*).

TRPV3 has a reported temperature threshold for activation between 31–39°C, with *Smith et al., 2002* noting activation at 39°C. Consistent with this, we observed the strongest physiological effects in cortical PNs at 39°C. At this temperature, intracellular blockade of TRPV3 reduced average spiking levels and temperature-induced increases in the PSP. Additionally, a potential early postnatal temperature-dependent activity mechanism in PNs, where levels of excitatory and inhibitory input are adjusted to match the cell's ST, was abolished with TRPV3 blockade. Altogether, these results suggest that TRPV3 blockade reduces net activity in sensory cortical circuits by decreasing activity in STAY cells.

Although TRPV3 channels are cation-nonselective, they exhibit high permeability to $Ca^{2+}$ ($Ca^{2+}>Na^+\approx K^+\approx Cs^+$) with permeability ratios (relative to $Na^+$) of 12.1, 0.9, 0.9, 0.9 (*Xu et al., 2002*). Opening of TRPV3 channels activates a nonselective cationic conductance and elevates membrane depolarization, which can increase the likelihood of generating action potentials. Indeed, our observations of a loss of the temperature-induced increases in the PSP with TRPV3 blockade are consistent with a reduction in membrane depolarization. In S1 cortical circuits at P12-14, STAY PNs appear to rely on a temperature-dependent activity mechanism, where depolarization levels (mediated by higher excitatory input and lower inhibitory input) are scaled to match the cell's ST. Thus, an inability to increase PSPs with temperature elevations prevents PNs from reaching ST, so they cease spiking. See summary *Figure 11*.

Previous research in *Trpv3* $^{-/-}$ KO mice revealed a similar phenotype of reduced activity with diminished TRPV3 function (*Chen et al., 2022*). Whole-cell current-clamp recordings from striatal MSNs and L2 stellate neurons in the entorhinal cortex at 36.5–37.5°C showed that *Trpv3*$^{-/-}$ cells exhibited reduced intrinsic excitability, as evidenced by fewer spikes in response to 400 ms current steps, a hyperpolarized RMP, and a decrease in mEPSC frequency (*Chen et al., 2022*). These electrophysiological effects were also replicated using pharmacological inhibition of TRPV3 with 50 μmol/L forsythoside B (*Chen et al., 2022*). Additionally, consistent with the idea that ischemic injury upregulates TRPV3 expression, leading to intracellular $Ca^{2+}$ overload and progressive cell death, inhibition of TRPV3, rather than N-methyl-D-aspartate (NMDA) receptors, protected against progressive cell death after stroke (*Chen et al., 2022*).

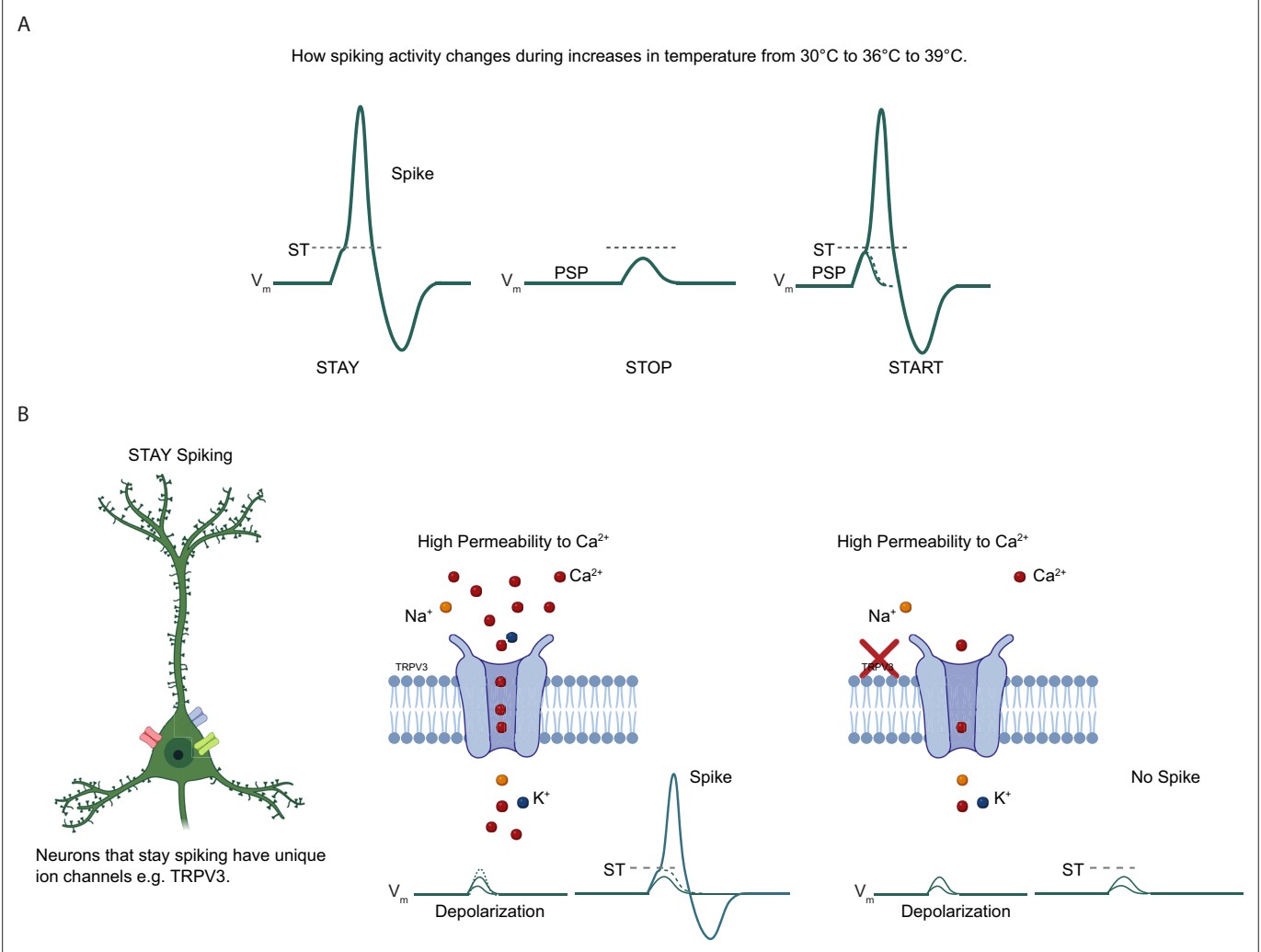

**Figure 11.** Summary model of how STAY neurons achieve firing stability through unique ion channel compositions, including TRPV3. (**A**) In cortical L2/3 pyramidal neurons (PNs) with synaptically evoked spiking, gradual increases in brain slice temperature from 30 °C to 36°C to 39°C result in four possible outcomes: neurons remain inactive, continue spiking (STAY), stop spiking, or initiate spiking. To spike, PNs must reach the spike threshold (ST), defined as the minimal $V_m$ that elicits an action potential, which requires sufficient depolarization via the post-synaptic potential (PSP). STAY neurons consistently reach ST and continue spiking. Neurons that stop spiking fall below the level of depolarization required to reach ST, while neurons that initiate spiking achieve sufficient depolarization to newly reach ST. (**B**) STAY neurons may contain unique ion channels, such as TRPV3. TRPV3 channels are highly permeable to $Ca^{2+}$ ions, and $Ca^{2+}$ influx contributes to PN depolarization. The presence of TRPV3 facilitates greater ion entry, enabling depolarization sufficient to reach ST and sustain spiking, whereas its absence reduces depolarization to levels insufficient for spiking.

As STAY PNs and neurons that initiate spiking at fever temperature ('START' cells) are active at fever temperature, future research will focus on the molecular characterization of these cells in the cortex and elsewhere in the brain at P12-14 and other ages. Interestingly, neurons that initiate spiking at fever temperature are more prevalent at P20-23 than at the earlier ages examined. TRPV4 block caused a subset of PNs to initiate spiking at 39°C, suggesting that low TRPV4 expression could define the population of START cortical PNs, at least at P12-14.

Molecular characterization of STAY and START cell types may provide additional insights into their function within normal brain circuits. In response to specific experiences, selective subsets of neurons, known as 'engrams,' may become active to encode specific memories or behavioral responses (*Lee et al., 2021*). Alternatively, neurons that become active could initiate responses to specific stimuli, such as an increase in brain temperature. In this latter scenario, these neurons may be part of the brain's homeostatic response to temperature increases, while STAY neurons may play a critical role in facilitating ongoing (cognitive) functions during a fever. Future research into the molecular characterization

of these cell types could offer opportunities to target and modulate their activity in a brain region-specific manner, potentially advancing treatments for fever-associated seizures or epilepsies, such as Dravet Syndrome (a genetic epilepsy typically initiating in early infancy) and febrile infection-related epilepsy syndrome (FIRES).

# Materials and methods

## Animals

All experimental procedures were performed in accordance with the National Institutes of Health guidelines for care and use of laboratory animals and the EC Council Directive of September 22, 2010 (2010/63/EU). All experiments were approved by the Animal Care and Use Committee of the National Institute on Alcohol Abuse and Alcoholism (NIAAA) (Protocol # LIN-MA-1) and the Animal Care Committee of the HUN-REN Research Centre for Natural Sciences (RCNS) and by the National Food Chain Safety Office of Hungary (license number: PE/EA/1004-5/2021). Mice were housed under standard conditions with ad libitum access to food and water, under a 12 hr light/dark cycle. The following mouse strains were obtained from The Jackson Laboratory (JAX): C57BL/6J (JAX 000664), FVB/N mice (JAX 004828), and Trpv3 (JAX 010773) (Moussaieff A et al., 2008; *Moqrich et al., 2005*). The day of birth was denoted as postnatal day (P)1. Both male and female mice were used for body temperature ($T_b$) recordings and electrophysiological experiments.

## Body temperature measurements

All mice used for body temperature ($T_b$) recordings were implanted with sterile IPTT-300 Implantable Programmable Temperature transponders (Bio Medic Data Systems, LLC) at 2 weeks of age. The injection site was pre-cleaned with a betadine solution, and the sterile transponders were injected subcutaneously using a pre-loaded 12-gauge syringe into animals anesthetized with 4–5% isoflurane. The injection site was on the left side, approximately 11 mm from the base of the hip, with the animal in the prone position. Post-implantation, the mice were able to move, eat, and drink autonomously, with no adverse phenotypes noted. $T_b$ recording commenced at least 5 days after injection. Prior to recording, mice were brought into the procedure room and allowed to habituate for 1 hr. $T_b$ was recorded every 5 min for 6 hr using an IPTT-300 Implantable Programmable Temperature reader (DAS-8027) in a 37L x 16W x 13H cm cage. $T_b$ was recorded in the presence (*Figure 1D*) or absence (*Figure 1A–C*) of infrared light. Infrared light exposure was provided via a 250-W temperature-controlled infrared lamp (catalog #50320, Stoelting), set to 32.5–33.5°C. The lamp was positioned ~15 cm above the cage, and the ambient temperature was monitored every 5 min (*Figure 1D*).

## Slice preparation

Using standard methods, acute primary somatosensory cortex (S1) slices (350 µm thick) from P7-8, P12-14, or P20-23 mice were cut in the 'across-row' plane, oriented 35° toward coronal from midsagittal (*Antoine et al., 2019*). The cutting solution contained (in mM): 85 NaCl, 75 sucrose, 25 D- (+)-glucose, 4 $MgSO_4$, 2.5 KCl, 1.25 $NaH_2PO_4$, 0.5 ascorbic acid, 25 $NaHCO_3$, 0.5 $CaCl_2$. Once cut, slices were transferred to a submerged-style holding chamber containing standard Ringer's solution (in mM: 119 NaCl, 2.5 KCl, 1.3 $MgSO_4$, 1 $NaH_2PO_4$, 26.2 $NaHCO_3$, 11 D-(+)-glucose, and 2.5 $CaCl_2$) and then incubated at 32°C for 30 min. Both solutions were at neural pH (i.e. 7.3), 300 mOsm, and saturated with 95% $O_2$ and 5% $CO_2$. Slices were kept at room temperature for at least 30 min before being transferred to a submerged recording chamber.

## In vitro physiology

Whole-cell current-clamp recordings were made from PNs visually identified via infrared DIC optics. Physiological verification for regular spiking was done in current clamp. Recordings were made using 3–6 MΩ micropipettes containing (in mM): 116 K gluconate, 20 HEPES, 6 KCl, 2 NaCl, 0.5 EGTA, 4 MgATP, 0.3 NaGTP, 10 Na phosphocreatine, with a Multiclamp 700B amplifier (Molecular Devices, Sunnyvale, CA). Where applicable, forsythoside B (50 µM, TRPV3 blocker, Millipore Sigma Cat #: PHL83313) or RN1734 (10 µM, *TRPV4 blocker*, Tocris Bioscience Cat #: 3746/25) was added to the internal solution. Camphor (MedChem, Cat. No: HY-N0808) was added to the bath solution for experiments in *Figure 8*.

Signals were filtered (2–6 kHz) and digitized (10–20 kHz). Perfusate temperature for all in vitro experiments was regulated by a PC-connected Peltier heater (SM-4600, Scientifica, UK) and temperature controller (Linlab2, Scientifica, UK), with a high-accuracy, low-noise temperature control system, containing both a built-in temperature sensor and bath sensor for accurate feedback. ACSF flow rates of ~3 ml/min were used to facilitate efficient heating/cooling.

To determine the effect of temperature on an activated cortical network, layer 4-evoked spiking was quantified during perfusate temperature increases from 30°C to 36°C, and then to fever range (~39°C) while recording membrane potential. The slicing plane facilitated clear identification of whisker barrel columns, and neuronal activity was evoked by stimulating the barrel center using a bipolar electrode (0.2 ms pulses) at a stimulation of 1.4 E$\theta$. E$\theta$ is defined as the minimal intensity that evoked a consistent excitatory post-synaptic current (EPSC) during more than 3 of 5 consecutive sweeps with 10 s ISI (*Antoine et al., 2019*). E$\theta$ was determined in voltage clamp for each recorded cell prior to recording PSPs and spiking in current clamp. A new brain slice was used for each recording. L4-evoked feedforward post-synaptic potentials (PSPs) and spiking were recorded in single L2/3 PNs from a pre-stimulus baseline membrane potential ($V_m$) of –50 mV. This $V_m$, just below spike threshold, was selected to mimic in vivo conditions, as PNs in vivo can reside within this $V_m$ range during whisker exploration of objects or surfaces (*Yamashita et al., 2013*). Evoked spikes were analyzed over a 150 ms interval, starting 2–3 ms post-stimulus, for 11 sweeps at 10 s inter-sweep interval (ISI). Spike threshold was defined as the membrane potential ($V_m$) at which the second derivative of $V_m$ was >5 SDs above the pre-stimulus period. Intrinsic spiking excitability was measured in the presence of glutamate and GABA-A receptor blockers (in µM: 100 APV, 10 NBQX, 3 gabazine). F-I curves were obtained from PNs held at –80 mV by applying incremental current injections relative to this $V_m$.

## Animal surgery and in vivo electrophysiological recordings

In vivo experiments were done under general anesthesia where five mice (age, P24-26; body weight, 7–10 g; both genders) received an intraperitoneal injection of ketamine (100 mg/kg) and xylazine (10 mg/kg). Regular doses of the ketamine/xylazine cocktail were given intramuscularly to maintain the depth of anesthesia during the experimental sessions. Up until the thermal fever protocol started, the $T_b$ of the animals was kept at 36°C using a homeothermic heating pad connected to a temperature controller (Supertech, Pécs, Hungary). To measure the internal body temperature, a Type T thermocouple microprobe (MT-29/5; Physitemp Instruments, Clifton, NJ, USA) was placed in the rectum of the animal. The microprobe has a shaft diameter of 330 µm, a time constant of 0.025 s, and 0.1°C accuracy.

To perform high-density extracellular electrophysiological recordings, mice were placed in a stereotaxic frame (David Kopf Instruments, Tujunga, CA, USA), then two circular craniotomies (~1.5 mm in diameter) were made with a dental drill above the left and right barrel cortices. The craniotomies were centered at the following stereotaxic coordinates: anterior-posterior (AP): –1.0 mm; medial-lateral (ML): 3.5 mm (with respect to the bregma; *Paxinos and Franklin, 2001*). Two commercially available Neuropixels 1.0 silicon probes (imec, Leuven, Belgium; *Jun et al., 2017*) mounted on two motorized stereotaxic micromanipulators (Neurostar, Tübingen, Germany) were implanted into the brain, one into the left and the other into the right barrel cortex, to a depth of 1.5 mm. An insertion speed of 2 µm/s was used to decrease the mechanical trauma caused by the probe insertion (*Fiáth et al., 2019*).

In order for the probe tracks to be perpendicular to cortical layers, the probes were inserted at an angle of 20 degrees from vertical. The dura mater in the craniotomy was left intact, except when it was too thick and thus the silicon probe could not pierce through this layer (which was indicated by significant probe buckling; n=3 insertions). In these cases, a 36-gauge, slightly bent needle was used to carefully cut the dura above the targeted cortical area. After the probe reached its final insertion depth, to allow the brain tissue to settle, we waited at least 10 min before electrophysiological recording was started. Spiking activity of cortical neurons (action potential band, 300–10,000 Hz) was recorded on 384 channels (768 channels in total for the two probes), with a sampling rate of 30 kHz/channel and with a gain of 500 (yielding a resolution of 2.34 µV per bit). Data were acquired using the SpikeGLX open-source software (http://billkarsh.github.io/SpikeGLX/; *Karsh, 2026*). A common stainless steel wire inserted into the neck muscle of the animal served as the external reference and ground electrode. To avoid the dehydration of the cortex, the skull and the craniotomy was kept moist during the whole experiment using body temperature, sterile physiological saline solution, and Gelaspon.

Before starting the experimental protocol, manual whisker stimulation (by repetitively touching the whiskers of the animal with a cotton swab) was used to verify the recording position. In all cases (n=10 probe insertions), we detected strong whisker-evoked neuronal activity.

## Thermal fever protocol during electrophysiological recordings

First, we recorded cortical activity for 45 min at ~36°C body temperature ('baseline'" period), then the body temperature of the animal was elevated with the aid of the heating pad and a power bank with warming capability which was placed next to the mouse. The body temperature was increased from 36°C to 39°C in about 5 min (~0.01°C/s). The elevated body temperature was maintained for 45 min ('thermal fever' period). Next, we turned off the heating until the $T_b$ reached physiological temperatures (36°C; ~5 min; ~0.01°C/s). Cortical activity was recorded for another 45 min at 36°C body temperature ('recovery' period). Continuous recordings with a total duration of 145 min were obtained for each mouse.

## Spike sorting and data analysis

To extract cortical single-unit activity, spike sorting was performed with Kilosort2 (https://github.com/MouseLand/Kilosort; *Pachitariu et al., 2016a*; *Pachitariu et al., 2016b*; *Pachitariu and Pennington, 2026*) using the default parameter set (available in the StandardConfig.m file). Channels containing activity acquired by recording sites located outside the cortex were removed before spike sorting (usually ~130 channels/probe recorded from the barrel cortex). The list of single-unit clusters generated by Kilosort2 was visually inspected to remove units considered as noise (e.g. units with abnormal spike waveform shapes) or multi-unit activity (e.g. clusters with a contaminated refractory period). Manual curation of the Kilosort2 results was done with the Phy Python library, which provides a graphical user interface for interactive visualization of high-density data and supply operations for merging, splitting, and marking of clusters (https://github.com/cortex-lab/phy; *Rossant et al., 2016*; *Rossant, 2026*). In this dataset, we aimed to keep only those single units which had at least 900 spikes (>~0.1 Hz firing rate), a clear refractory period, a consistent waveform shape, and whose spikes were present throughout the 145-min-long recording.

Following manual curation, based on their spike waveform duration, the selected single units (n=633) were separated into putative inhibitory interneurons and excitatory principal cells (*Barthó et al., 2004*). The spike duration was calculated as the time difference between the trough and the subsequent waveform peak of the mean filtered (300–6000 Hz bandpassed) spike waveform. Durations of extracellularly recorded spikes showed a bimodal distribution (Hartigan's dip test; $p<0.001$) characteristic of the neocortex with shorter durations corresponding to putative interneurons (narrow spikes) and longer durations to putative principal cells (wide spikes). Next, k-means clustering was used to separate the single units into these two groups, which resulted in 140 interneurons (spike duration <0.6 ms) and 493 principal cells (spike duration >0.6 ms), corresponding to a typical 22–78% (interneuron – principal) cell ratio. Finally, the firing rates of neurons were computed separately during the three distinguished periods (baseline, thermal fever, and recovery). To decrease the effect of transient changes (e.g. tissue recovery during the baseline period or the short-term effect of heating during the thermal fever period), for each 45-min-long period, we used only the last 25 min to calculate the mean firing rates of neurons.

## Immunofluorescence staining

Mice were anesthetized and transcardially perfused using standard methods. Brains were post-fixed overnight in 4% PFA at 4°C with shaking, rinsed in 1X PBS, and cryoprotected in 30% sucrose. They were embedded in optimal cutting temperature (OCT) compound, frozen on dry ice, and stored at –80°C until sectioning. Tissue was cut at 30 µm on a cryostat and mounted onto slides. Sections were permeabilized in 0.3% Triton X-100 (in 1X PBS) for 10 min at room temperature. Endogenous peroxidase activity was quenched by incubation in 3% $H_2O_2$ (in 1X PBS) at room temperature. Slides were blocked in 10% normal goat serum for 1 hr at room temperature, then incubated overnight at 4°C with anti-TRPV3-Biotin antibody (#ACC-033-B), TRPV3 blocking peptide (#BLP-CC033), and/or anti-TRPV4 antibody (#ACC-034) (Alomone Labs, Israel). For antibody–peptide controls, the anti-TRPV3-Biotin antibody was pre-incubated with its blocking peptide at a 1:2.5 ratio, using 1 mg/ml of each. The following day, sections were incubated for 1 hr at room temperature with fluorescently

labeled secondary antibodies (streptavidin-conjugated or unconjugated). Nuclei were counterstained with DAPI Fluoromount-G (Electron Microscopy Sciences, USA). Finally, sections were mounted onto slides, cover-slipped, and stored at 4°C until imaging. Images were acquired using a Keyence BZ-X810 microscope (BZ-X800 Viewer, Version 1.1.1.8) with BZ-X filter sets: DAPI (Ex 360/40 nm; OP-87762), GFP (Ex 470/40 nm; OP-87763), and TRITC (Ex 545/25 nm; OP-87764).

## Statistics

Statistical analyses were conducted using Prism 9.0 (GraphPad). For experiments, at least three mice from a minimum of two separate litters were used, with littermates typically included across the age groups P7-8, P12-14, and P20-23. Non-Gaussian data were either log-transformed for parametric testing or subjected to nonparametric tests, as specified in the results and figure legends. Significance was consistently reported at $\alpha=0.05$. Details on the statistical tests used are provided in the figure captions.

## Acknowledgements

This research was supported in part by the Intramural Research Program of the National Institutes of Health (NIH). The contributions of the NIH authors were made as part of their official duties as NIH federal employees, are in compliance with agency policy requirements, and are considered Works of the United States Government. However, the findings and conclusions presented in this paper are those of the authors and do not necessarily reflect the views of the NIH or the U.S. Department of Health and Human Services. MWA, RF, and YS designed, performed, interpreted, and analyzed physiology experiments. MWA and YS designed, performed, and analyzed ex vivo electrophysiological experiments. RF designed, performed, and analyzed in vivo electrophysiological experiments. BMK performed the immunohistochemistry for TRPV3 and TRPV4. MWA wrote the paper. MWA and IU provided project supervision. We thank Dr David Lovinger, Scientific Director of NIAAA, for his critical review of the manuscript. Research support was provided from the National Institute on Alcohol Abuse and Alcoholism (NIAAA) of the National Institutes of Health (NIH) under grant 1ZIAAA000440-02 (MWA), the National Institute of Neurological Disorders and Stroke under grant 1K22NS105922-01 (MWA), the Hungarian Brain Research Program Grant NAP2022-I-2/2022 (IU) and the Bolyai János Scholarship of the Hungarian Academy of Sciences (RF). Project no. 150574 has been implemented with the support provided by the Ministry of Culture and Innovation of Hungary from the National Research, Development, and Innovation Fund, financed under the STARTING_24 funding scheme.

## Additional information

### Competing interests

Michelle W Antoine: Reviewing editor, eLife. The other authors declare that no competing interests exist.

### Funding

| Funder | Grant reference number | Author |
| --- | --- | --- |
| National Institutes of Health | 1ZIAAA000440-02 | Michelle W Antoine |
| Hungarian Brain Research Program | NAP2022-I-2/2022 | István Ulbert |
| Hungarian Academy of Sciences | Bolyai Janos Scholarship | Richárd Fiáth |
| National Institutes of Health | 1K22NS105922-01 | Michelle W Antoine |

The funders had no role in study design, data collection and interpretation, or the decision to submit the work for publication.

## Author contributions
Yiming Shen, Formal analysis, Investigation; Richárd Fiáth, Formal analysis, Funding acquisition, Investigation, Methodology; Baskar Mohana Krishnan, Methodology; István Ulbert, Resources, Supervision, Funding acquisition; Michelle W Antoine, Conceptualization, Formal analysis, Supervision, Funding acquisition, Investigation, Visualization, Methodology, Writing – original draft, Writing – review and editing

## Author ORCIDs
Michelle W Antoine ⬤ https://orcid.org/0000-0001-7222-7309

## Ethics
All experimental procedures were performed in accordance with the National Institutes of Health guidelines for care and use of laboratory animals and the EC Council Directive of September 22, 2010 (2010/63/EU). All experiments were approved by the Animal Care and Use Committee of the National Institute on Alcohol Abuse and Alcoholism (NIAAA) (Protocol # LIN-MA-1) and the Animal Care Committee of the HUN-REN Research Centre for Natural Sciences (RCNS) and by the National Food Chain Safety Office of Hungary (license number: PE/EA/1004- 5/2021).

Reviewer #1 (Public review): https://doi.org/10.7554/eLife.102412.3.sa1
Reviewer #2 (Public review): https://doi.org/10.7554/eLife.102412.3.sa2
Reviewer #3 (Public review): https://doi.org/10.7554/eLife.102412.3.sa3
Author response https://doi.org/10.7554/eLife.102412.3.sa4

# Additional files

## Supplementary files
MDAR checklist

## Data availability
Data associated with this manuscript can be found here: https://doi.org/10.5061/dryad.hhmgqnkwc.

The following dataset was generated:

| Author(s) | Year | Dataset title | Dataset URL | Database and Identifier |
| --- | --- | --- | --- | --- |
| Shen Y, Fiath R, Antoine MW | 2026 | TRPV3 channel activity helps cortical neurons stay active during fever | https://doi.org/10.5061/dryad.hhmgqnkwc | Dryad Digital Repository, 10.5061/dryad.hhmgqnkwc |

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
