## [Editor Report · eLife Assessment]

This is a **valuable** study of the physiological mechanisms promoting network activity during fever in the mouse neocortex. The supporting evidence is **solid**, and has improved with revision, along with increased clarity of presentation.

---

## [Referee Report · Reviewer #1 (Public review)]

The paper by Chen et al describes the role of neuronal themo-TRPV3 channels in the firing of cortical neurons at fever temperature range. The authors began by demonstrating that exposure to infrared light increasing ambient temperature causes body temperature rise to fever level above 38{degree sign}C. Subsequently, they showed that at the fever temperature of 39{degree sign}C, the increased spike threshold (ST) increased in both populations (P12-14 and P7-8) of cortical excitatory pyramidal neurons (PNs). However, the spike number only decreased in P7-8 PNs, while it remained stable in P12-14 PNs at 39{degree sign}C. In addition, the fever temperature also reduced the late peak postsynaptic potential (PSP) in P12-14 PNs. The authors further characterized the firing properties of cortical P12-14 PNs, identifying two types: STAY PNs that retained spiking at 30{degree sign}C, 36{degree sign}C and 39{degree sign}C, and STOP PNs that stopped spiking upon temperature change. They further extended their and analysis and characterization to striatal medium spiny neurons (MSNs) and found that STAY MSNs and PNs shared same ST temperature sensitivity. Using small molecule tools, they further identified that themo-TRPV3 currents in cortical PNs increased in response to temperature elevation, but not TRPV4 currents. The authors concluded that during fever, neuronal firing stability is largely maintained by sensory STAY PNs and MSNs that express functional TRPV3 channels. Overall, this study is well designed and executed with substantial controls, some interesting findings and quality of data.

Comments on revisions:

My previous concerns have been addressed in this revised manuscript.

---

## [Referee Report · Reviewer #2 (Public review)]

Summary:

The authors studied the excitability of layer 2/3 pyramidal neurons in response to layer four stimulation at temperatures ranging from 30 to 39{degree sign}C in P7-8, P12-P14, and P22-P24 animals. They also measure brain temperature and spiking in vivo in response to externally applied heat. Some pyramidal neurons continue to fire action potentials in response to stimulation at 39{degree sign}C and are referred to as "stay neurons." Stay neurons have unique properties, aided by the expression of the TRPV3 channel.

Strengths:

The authors focused on layer 2/3 neuronal excitability at three developmental stages: during the window of susceptibility to febrile seizures, before the window opens, and after it closes.

Electrophysiological experiments are rigorously performed and carefully interpreted.

The cellular electrophysiology is further confirmed. The authors compared the seizure susceptibility of TRPV3 knockout, heterozygous, and wild-type mice. EEG recording would have strengthened the study, but they are challenging in this age group.

Finally, the authors studied TRPV3 expression with immunohistochemistry.

---

## [Referee Report · Reviewer #3 (Public review)]

Summary:

This important study combines in vitro and in vivo recording to determine how the firing of cortical and striatal neurons changes during a fever range temperature rise (37-40 oC). The authors found that certain neurons will start, stop, or maintain firing during these body temperature changes. The authors further suggested that the TRPV3 channel plays a role in maintaining cortical activity during fever.

Strengths:

The topic of how the firing pattern of neurons changes during fever is unique and interesting. The authors carefully used in vitro electrophysiology assays to study this interesting topic.

Weaknesses:

(1) In vivo recording is a strength of this study. However, data from in vivo recording is only shown in Fig 5A,B. This reviewer suggests the authors further expand on the analysis of the in vivo Neuropixels recording. For example, to show single spike waveforms and raster plots to provide more information on the recording. The authors can also separate the recording based on brain regions (cortex vs striatum) using the depth of the probe as a landmark to study the specific firing of cortical neurons and striatal neurons. It is also possible to use published parameters to separate the recording based on spike waveform to identify regular principal neurons vs fast-spiking interneurons. Since the authors studied E/I balance in brain slices, it would be very interesting to see whether the "E/I balance" based on the firing of excitatory neurons vs fast-spiking interneurons might be changed or not in the in vivo condition.

(2) The author should propose a potential mechanism for how TRPV3 helps to maintain cortical activity during fever. Would calcium influx-mediated change of membrane potential be the possible reason? Making a summary figure to put all the findings into perspective and propose a possible mechanism would also be appreciated.

(3) The author studied P7-8, P12-14, and P20-26 mice. How do these ages correspond to the human ages? it would be nice to provide a comparison to help the reader understand the context better.

Comments on revisions:

In this revised version, the authors nicely addressed my critiques. I have no more comments to make.

---

## [Author Response]

The following is the authors’ response to the original reviews

**Public Reviews:**

**Reviewer #1 (Public review):**
The paper by Chen et al describes the role of neuronal themo-TRPV3 channels in the firing of cortical neurons at a fever temperature range. The authors began by demonstrating that exposure to infrared light increasing ambient temperature causes body temperature to rise to a fever level above 38{degree sign}C. Subsequently, they showed that at the fever temperature of 39{degree sign}C, the spike threshold (ST) increased in both populations (P12-14 and P7-8) of cortical excitatory pyramidal neurons (PNs). However, the spike number only decreased in P7-8 PNs, while it remained stable in P12-14 PNs at 39 degrees centigrade. In addition, the fever temperature also reduced the late peak postsynaptic potential (PSP) in P12-14 PNs. The authors further characterized the firing properties of cortical P12-14 PNs, identifying two types: STAY PNs that retained spiking at 30{degree sign}C, 36{degree sign}C, and 39{degree sign}C, and STOP PNs that stopped spiking upon temperature change. They further extended their analysis and characterization to striatal medium spiny neurons (MSNs) and found that STAY MSNs and PNs shared the same ST temperature sensitivity. Using small molecule tools, they further identified that themo-TRPV3 currents in cortical PNs increased in response to temperature elevation, but not TRPV4 currents. The authors concluded that during fever, neuronal firing stability is largely maintained by sensory STAY PNs and MSNs that express functional TRPV3 channels. Overall, this study is well designed and executed with substantial controls, some interesting findings, and quality of data. Here are some specific comments:(1) Could the authors discuss, or is there any evidence of, changes in TRPV3 expression levels in the brain during the postnatal 1-4 week age range in mice?

This is an excellent question. To our knowledge, no published studies have documented changes in TRPV3 expression in the mouse brain during the first to fourth postnatal weeks. Research on TRPV3 expression has primarily relied on RT-PCR analysis of RNA from dissociated adult brain tissue (Jang et al., 2012; Kumar et al., 2018), largely due to the limited availability of effective antibodies for brain sections at the time. Furthermore, the Allen Brain Atlas does not provide data on TRPV3 expression in the developing or postnatal brain. To address this gap, we performed immunohistochemistry to examine TRPV3 expression at P7,

P14, and P21 (Figure 7). To confirm specificity, the TRPV3 antibody was co-incubated with a TRPV3 blocker (Figure 7A, top row, right panel). While immunohistochemistry is semiquantitative, we observed a trend toward increased TRPV3 expression in the cortex, striatum, hippocampus, and thalamus from P7 to P14.

(2) Are there any differential differences in TRPV3 expression patterns that could explain the different firing properties in response to fever temperature between the STAY- and STOP neurons?

This is another excellent question, and we plan to explore it in the future by developing reporter mice for TRPV3 expression and viral tools that leverage endogenous TRPV3 promoters to drive a fluorescent protein, enabling monitoring of cells with native TRPV3 expression. To our knowledge, such tools do not currently exist. Creating them will be challenging, as it requires identifying promoters that accurately reflect endogenous TRPV3 expression.

We have not yet quantified TRPV3 expression in STOP and STAY neurons. However, our analysis of evoked spiking at 30, 36, and 39 °C suggests that TRPV3 may mark a population of cortical pyramidal neurons that tend to remain active (“STAY”) as temperatures increase. While we have not directly compared TRPV3 expression between STAY and STOP neurons at feverrange temperatures, intracellular blockade of TRPV3 with forsythoside B (50 µM) significantly reduced the proportion of STAY neurons (Figure 9B). Consistently, spiking was also significantly reduced in Trpv3⁻/⁻ mice (Figure 10D).

In our immunohistochemical analysis, TRPV3 was detected in L4 barrels and in L2/3, where we observed a patchy distribution with some regions showing more intense staining (Figure 7B). It is possible that cells with higher TRPV3 levels correspond to STAY neurons, while those with lower levels correspond to STOP neurons. As we develop tools to monitor activity based on endogenous TRPV3 levels, we anticipate gaining deeper insight into this relationship.

(3) TRPV3 and TRPV4 can co-assemble to form heterotetrameric channels with distinct functional properties. Do STOP neurons exhibit any firing behaviors that could be attributed to the variable TRPV3/4 assembly ratio?

There is some evidence that TRPV3 and TRPV4 proteins can physically associate in HEK293 cells and native skin tissues (Hu et al., 2022).TRPV3 and TRPV4 are both expressed in the cortex (Kumar et al., 2018), but it remains unclear whether they are co-expressed and coassembled to form heteromeric channels in cortical excitatory pyramidal neurons. Examination of the I-V curve from HEK cells co-expressing TRPV3/4 heteromeric channels shows enhanced current at negative membrane potentials (Hu et al., 2022).

Currently, we cannot characterize cells as STOP or STAY and measure TRPV3 or TRPV4 currents simultaneously, as this would require different experimental setups and internal solutions. Additionally, the protocol involves a sequence of recordings at 30, 36, and 39°C, followed by cooling back to 30°C and re-heating to each temperature. Cells undergoing such a protocol will likely not survive till the end.

In our recordings of *TRPV3* currents, which likely include both STOP and STAY cells, we do not observe a significant current at negative voltages, suggesting that *TRPV3/4* heteromeric channels may either be absent or underrepresented, at least at a 1:1 ratio. However, the possibility that TRPV3/4 heteromeric channels could define the STOP cell population is intriguing and plausible.

(4) In Figure 7, have the authors observed an increase of TRPV3 currents in MSNs in response to temperature elevation?

We have not recorded TRPV3 currents in MSNs in response to elevated temperatures. Please note that the handling editor gave us the option to remove these data from the paper, and we elected to do so to develop them as a separate manuscript.

(5) Is there any evidence of a relationship between TRPV3 expression levels in D2+ MSNs and degeneration of dopamine-producing neurons?

This is an interesting question, though it falls outside our current research focus in the lab. A PubMed search yields no results connecting the terms TRPV3, MSNs, and degeneration. However, gain-of-function mutations in TRPV4 channel activity have been implicated in motor neuron degeneration (Sullivan et al., 2024) and axon degeneration (Woolums et al., 2020). Similarly, TRPV1 activation has been linked to developmental axon degeneration (Johnstone et al., 2019), while TRPV3 blockade has shown neuroprotective effects in models of cerebral ischemia/reperfusion injury in mice (Chen et al., 2022).

The link between TRPV activation and cell degeneration, however, may not be straightforward. For instance, TRPV1 loss has been shown to accelerate stress-induced degradation of axonal transport from retinal ganglion cells to the superior colliculus and to cause degeneration of axons in the optic nerve (Ward et al., 2014). Meanwhile, TRPV1 activation by capsaicin preserves the survival and function of nigrostriatal dopamine neurons in the MPTP mouse model of Parkinson's disease (Chung et al., 2017).

(6) Does fever range temperature alter the expressions of other neuronal Kv channels known to regulate the firing threshold?

This is an active line of investigation in our lab. The results of ongoing experiments will provide further insight into this question.

**Reviewer #2 (Public review):**
Summary:The authors study the excitability of layer 2/3 pyramidal neurons in response to layer four stimulation at temperatures ranging from 30 to 39 Celsius in P7-8, P12-P14, and P22-P24 animals. They also measure brain temperature and spiking in vivo in response to externally applied heat. Some pyramidal neurons continue to fire action potentials in response to stimulation at 39 C and are called stay neurons. Stay neurons have unique properties aided by TRPV3 channel expression.Strengths:The authors use various techniques and assemble large amounts of data.Weaknesses:(1) No hyperthermia-induced seizures were recorded in the study.

The goal of this manuscript is to uncover age-related physiological changes that enable the brain to maintain function at fever-range temperatures, typically 38–40°C. Febrile seizures in humans are also typically induced within this temperature range. Given this context, we initially did not examine hyperthermia-induced seizures. However, as requested, we assessed the effects of reduced *Trpv3* expression on hyperthermia-induced seizures in WT(*Trpv3+/+*), heterozygous (*Trpv3+/-*), and homozygous knockout (*Trpv3-/-*) P12 pups. Please see figure 10.

While T_b_ at seizure onset and the rate of T_b_ increase leading to seizure were not significantly different among genotypes, the time to seizure from the point of loss of postural control (LPC), defined as collapse and failure to maintain upright posture, was significantly longer in *Trpv3+/-* and *Trpv3-/-* mice. Together, these results indicate that reduced TRPV3 function enhances resistance to seizure initiation and/or propagation under febrile conditions, likely by decreasing neuronal depolarization and excitability.

(2) Febrile seizures in humans are age-specific, extending from 6 months to 6 years. While translating to rodents is challenging, according to published literature (see Baram), rodents aged P11-16 experience seizures upon exposure to hyperthermia. The rationale for publishing data on P7-8 and P22-24 animals, which are outside this age window, must be clearly explained to address a potential weakness in the study.

As requested, we have added an explanation in the “Introduction” for our rationale in including age ranges that flank the period of susceptibility to hyperthermia-induced seizures (see lines 80–100). In summary, we emphasize that this design provides negative controls, allowing us to determine whether the changes observed in the P12–14 window are specific to this developmental period.

(3) Authors evoked responses from layer 4 and recorded postsynaptic potentials, which then caused action potentials in layer 2/3 neurons in the current clamp. The post-synaptic potentials are exquisitely temperature-sensitive, as the authors demonstrate in Figures 3 B and 7D. Note markedly altered decay of synaptic potentials with rising temperature in these traces. The altered decays will likely change the activation and inactivation of voltage-gated ion channels, adjusting the action potential threshold.

The activation and inactivation of voltage-gated ion channels can modulate action potential threshold. Indeed, we have identified channels that contribute to the temperature-induced increase in spike threshold, including BK channels and *Scn2a*. However, Figure 4B represents a cell with no inhibition at 39°C, and thus the observed loss of the late postsynaptic potential (PSP). This primarily contributes to the prolonged decay of the synaptic potentials. By contrast, cells in which inhibition is retained, when exposed to the same thermal protocol, do not exhibit such extended decay.

(4) The data weakly supports the claim that the E-I balance is unchanged at higher temperatures. Synaptic transmission is exquisitely temperature-sensitive due to the many proteins and enzymes involved. A comprehensive analysis of spontaneous synaptic current amplitude, decay, and frequency is crucial to fully understand the effects of temperature on synaptic transmission.

We did not intend to imply that E-I balance is generally unchanged at higher temperatures. Our statements specifically referred to observations in experiments conducted during the P20–26 age range in cortical pyramidal neurons. We are conducting a parallel line of investigation examining the differential susceptibility of E-I balance across age and temperature, and we have observed age- and temperature-dependent effects. Recognizing that our earlier wording may have been misleading, we have removed this statement from the manuscript.

(5) It is unclear how the temperature sensitivity of medium spiny neurons is relevant to febrile seizures. Furthermore, the most relevant neurons are hippocampal neurons since the best evidence from human and rodent studies is that febrile seizures involve the hippocampus.

Thank you for the opportunity to provide clarification. The goal of this manuscript is to uncover age-related physiological changes that enable the brain to maintain stable, non-excessive neuronal firing at fever-range temperatures (typically 38–40°C). We hypothesize that these changes are a normal part of brain development, potentially explaining why most children do not experience febrile seizures. By understanding these mechanisms, we may identify points in the process that are susceptible to dysfunction, due to genetic mutations, developmental delays, or environmental factors, which could provide insight into the rare cases when seizures occur between 2–5 years of age.

Our aim was not to establish a link between medium spiny neuron (MSN) function and febrile seizures. MSNs were included in this study as a mechanistic comparison because they represent a non-pyramidal, non-excitatory neuronal subtype, allowing us to assess whether the physiological changes observed in L2/3 excitatory pyramidal neurons are unique to these cells. Please note that the handling editor gave us the option to remove these data from the manuscript, and we chose to do so, developing these findings into a separate manuscript.

(6) TRP3V3 data would be convincing if the knockout animals did not have febrile seizures.

We find that approximately equal numbers of excitatory neurons either start or stop firing at fever-range temperatures (typically 38–40 °C). Neurons that continue to fire (“STAY” cells), thus play a key role in maintaining stable, non-excessive network activity. While future studies will examine the mechanisms driving some neurons to initiate spiking, our findings suggest that a reduction in the number of STAY cells could influence more subtle aspects of seizure dynamics, such as time to onset, by decreasing overall network excitability. We assessed the effects of reduced *Trpv3* expression on hyperthermia-induced seizures in WT(*Trpv3+/+*), heterozygous (*Trpv3+/-*), and homozygous knockout (*Trpv3-/-*) P12 pups. As you stated, these mice have hyperthermic seizures, however, we noted that the time to seizure from the point of loss of postural control (LPC), defined as collapse and failure to maintain upright posture, was significantly longer in *Trpv3+/-* and *Trpv3-/-* mice. Normally, seizures happen shortly after this point, but notably, *Trpv3-/-* mice took twice as long to reach seizure onset compared with wildtype mice. In an epileptic patient, this increased time may be sufficient for a caretaker to move the patient to a safer location, reducing the risk of injury during the seizure.

Consistent with findings that TRPV3 blockade using 50 µM forsythoside B reduces spiking in cortical L2/3 pyramidal neurons, we observed significantly reduced spiking in *Trpv3-/-* mice as well (Figure 10D). Analysis of postsynaptic potentials in these neurons showed that, in WT mice, PSP amplitude increased with temperature elevation into the febrile range, whereas this temperature-dependent depolarization was absent in *Trpv3-/-* mice (Figure 10E). Together, these results indicate that reduced TRPV3 function enhances resistance to seizure initiation and/or propagation under febrile conditions, likely by decreasing neuronal depolarization and excitability.

**Reviewer #3 (Public review):**
Summary:This important study combines in vitro and in vivo recording to determine how the firing of cortical and striatal neurons changes during a fever range temperature rise (37-40 oC). The authors found that certain neurons will start, stop, or maintain firing during these body temperature changes. The authors further suggested that the TRPV3 channel plays a role in maintaining cortical activity during fever.Strengths:The topic of how the firing pattern of neurons changes during fever is unique and interesting. The authors carefully used in vitro electrophysiology assays to study this interesting topic.Weaknesses:(1) In vivo recording is a strength of this study. However, data from in vivo recording is only shown in Figures 5A,B. This reviewer suggests the authors further expand on the analysis of the in vivo Neuropixels recording. For example, to show single spike waveforms and raster plots to provide more information on the recording. The authors can also separate the recording based on brain regions (cortex vs striatum) using the depth of the probe as a landmark to study the specific firing of cortical neurons and striatal neurons. It is also possible to use published parameters to separate the recording based on spike waveform to identify regular principal neurons vs fast-spiking interneurons. Since the authors studied E/I balance in brain slices, it would be very interesting to see whether the "E/I balance" based on the firing of excitatory neurons vs fast-spiking interneurons might be changed or not in the in vivo condition.

As requested, we have included additional analyses and figures related to the in vivo recording experiments in Figure 5. Specifically, we added examples of multiunit and single-spike waveforms, as well as autocorrelation histograms (ACHs). ACHs were used because raster plots of individual single units would not be very informative given the long recording period. Additionally, Figure 5F was also aimed to replace raster plots as it helps to track changes in the firing rate of a single neurons over time.

Additionally, all recordings were conducted in the cortex at a depth of ~1 mm from the surface, and no recordings were performed in the striatum. Based on the reviewing editor’s suggestions, we decided to remove the striatal data from the manuscript and develop this aspect of the project for a separate publication.

Lastly, we used published parameters to classify recordings based on spike waveform into putative regular principal neurons and interneurons. To clarify this point, we have now included descriptions that were previously listed only in the “Methods” section into the “Results” section as well.

The paragraph below from the methods section describes this procedure.

“Following manual curation, based on their spike waveform duration, the selected single units (n = 633) were separated into putative inhibitory interneurons and excitatory principal cells (Barthóet al., 2004). The spike duration was calculated as the time difference between the trough and the subsequent waveform peak of the mean filtered (300 – 6000 Hz bandpassed) spike waveform. Durations of extracellularly recorded spikes showed a bimodal distribution (Hartigan’s dip test; p < 0.001) characteristic of the neocortex with shorter durations corresponding to putative interneurons (narrow spikes) and longer durations to putative principal cells (wide spikes). Next, k-means clustering was used to separate the single units into these two groups, which resulted in 140 interneurons (spike duration < 0.6 ms) and 493 principal cells (spike duration > 0.6 ms), corresponding to a typical 22% - 78% (interneuron – principal) cell ratio”.

As suggested, we calculated the E/I balance using the average firing rates of excitatory and inhibitory neurons in the in vivo condition. Our analysis revealed that the E/I balance remained unchanged (see Author response image 1). Nonetheless, following the option provided by the reviewing editor, we have chosen to remove the statement referencing E/I balance from the manuscript.

**Author response image 1. sa4fig1:** 

(2) The author should propose a potential mechanism for how TRPV3 helps to maintain cortical activity during fever. Would calcium influx-mediated change of membrane potential be the possible reason? Making a summary figure to put all the findings into perspective and propose a possible mechanism would also be appreciated.

Thank you for your helpful suggestion. In response, we have included a summary figure (Figure 11) illustrating the hypothesis described in the Discussion section. We agree with your assessment that *Trpv3* most likely contributes to maintaining cortical activity during fever by promoting calcium influx and depolarizing the membrane potential.

(3) The author studied P7-8, P12-14, and P20-26 mice. How do these ages correspond to the human ages? it would be nice to provide a comparison to help the reader understand the context better.

Ideally, the mouse to human age comparison should depend on the specific process being studied. Per your suggestion, we have added additional references in the Introduction (Dobbing and Sands, 1973; Baram et al., 1997; Bender et al., 2004) to help readers better understand the correspondence between mouse and human ages.

**Recommendations for the authors:**

**Reviewer #2 (Recommendations for the authors):**
(3) Perform I-F curves to study the intrinsic properties of layer 2/3 neurons without the confound of evoked responses.

We performed F-I curve analyses (Figures 2H–I), as suggested by Reviewer 2, to study intrinsic properties of L2/3 neurons without evoked responses. Although rheobase increased at 39 °C compared to 30 °C, consistent with findings such as depolarized spike threshold and reduced input resistance, the mean number of spikes across current steps did not differ.

**Reviewer #3 (Recommendations for the authors):**
Some statistical descriptions are not clearly stated. For example, what statistical methods were used in Fig 2E? The effect size in Fig 2D seems to be quite small. The authors are advised to consider "nested analysis" to further increase the rigor of the analysis. Does each dot mean one neuron? Some of the data points might not be totally independent. The author should carefully check all figures to make sure the stats methods are provided for each panel.

We apologize for not including statistical details in Figure 2E. We have now added this information and verified that statistical descriptions are provided in all figure legends. In Figure 2D, each dot represents a cell, with measurements taken from the same cell at 30°C, 36°C, and 39°C. Given this design, the appropriate test is a one-way repeated-measures ANOVA.